# Catastrophic famine in Gaza: Unprecedented levels of hunger post-October 7th. A real population-based study from the Gaza Strip

MoezAllslam Faris [1☉*], Ayman S. Abutair[2☉*], Reham M. Elfarra[3], Nida. A. Barqawi[4], Amal M. Firwana[5], Rawan M. Firwana[5], Madleen M. AbuHajjaj[6], Shaimaa A. Shamaly[2], Samar S. AbuSamra[2], Hanan S. Bashir[6], Noor A. Abedalrahim[2], Noor A. Nofal[5], Mhran K. Alshawaf[7], Rania M. Al Shatali[2], Kafa I. Ghaben[5], Moayad I. Alron[8], Sara S. Alqeeq[5], Aya O. Al-Nabahin[2], Reem A. Badawi[6]

1 Departmen of Clinical Nutrition and Dietetics, Faculty of Allied Medical Sciences, Applied Science Private University, Amman, Jordan, 2 Clinical Nutrition and Dietitian Department, Al-Azhar University-Gaza, Gaza, Palestine, 3 UNRWA- Gaza, Gaza, Palestine, 4 Indepenedent Researcher, Amman, Jordan, 5 Health and Nutrition Department, University of Palestine, Gaza, Palestine, 6 Palestine Technical College-Gaza, Gaza, Palestine, 7 Ministry of Health, Nablus, Palestine, 8 Department Health Management, Islamic University-Gaza, Gaza, Palestine

☉ Equal contribution.
* moezalislam@gmail.com (MF); as.abutair@gmail.com (ASA)

## Abstract

### Background

The Gaza Strip, spanning approximately 365 square kilometers, has been a focal point of geopolitical tensions and humanitarian crises. The military escalation on October 7th exacerbated existing vulnerabilities, notably food security and hunger, with an estimated 85750 deaths due to Israeli attacks, representing about 8% of the 2.34 million population. This research aims to provide policymakers and humanitarian organizations with actionable insights, such as identifying the most vulnerable populations, quantifying the impact of specific restrictions, and informing the development and implementation of targeted interventions that improve long-term food security and alleviate human suffering in Gaza.

### Methods

A cross-sectional study was conducted from May to July 2024, assessing food insecurity and hunger among Palestinian households across the five governorates of Gaza. The study applied a quantitative research approach, utilized the Household Food Security Survey Module (HFSSM), Household Food Insecurity Access Scale (HFIAS), and Household Hunger Scale (HHS) to measure food insecurity, famine, and hunger. Self-reported anthropometric data and socioeconomic status were also collected. Data were analyzed using SPSS version 29, employing correlation tests, chi-square analysis, and logistic regression.

**Data availability statement:** The data underlying the study's results are available from the following URL: https://figshare.com/s/489ea0f86d08642374f0 (DOI: 10.6084/m9.figshare.28500518).

**Funding:** The author(s) received no specific funding for this work.

**Competing interests:** The authors have declared that no competing interests exist.

## Results

A survey of 1209 households across the Gaza Strip revealed a catastrophic humanitarian crisis. More than 54% of households experienced complete house destruction. Food insecurity reached unprecedented levels, with about 98% of households experiencing severe food insecurity, according to the HFIAS, while 100% experienced different levels of food insecurity as per the HFSSM. A staggering 95% of households experienced other sorts of hunger. The war was associated with significant ($p < 0.001$) weight loss among individuals, with the average weight dropping from $74.8 \pm 15.9$ kg before the war to $64.8 \pm 15.2$ kg, concomitant with significant ($p < 0.001$) reduction in BMI from $26.4 \pm 5.4$ to $22.8 \pm 5.2$ kg/m$^2$. Factors such as displacement, age, socioeconomic status, and educational level significantly exacerbated hunger severity.

## Conclusion

The study reveals a severe food insecurity and hunger crisis in the Gaza Strip, exacerbated by the ongoing damaging attacks by Israeli forces. These findings underscore the urgent need for immediate and sustained humanitarian assistance to address the critical food security and nutritional needs of the Gazan population.

## Introduction

The Gaza Strip, a small territory of approximately 365 square kilometers, has long been a focal point of geopolitical clashes, economic challenges, and humanitarian crises. Following the escalation of military actions on October 7th, the region has experienced heightened levels of Israeli forces attacks, resulting in significant disruptions to infrastructure, everyday life, and essential services. This has further exacerbated existing vulnerabilities, particularly concerning food security and the prevalence of hunger among the Gazan population. As a sequela, it has been estimated that about 85,750 deaths resulted only in the Gaza Strip caused by Israeli attacks, accounting for about 8% of the 2.34 million population [1]. This horrible killing rate emphasizes the urgent need for a ceasefire and humanitarian aid, highlighting the critical importance of documenting the full extent of the tragedy for historical accountability and future recovery efforts.

Hunger and food insecurity are critical public health concerns in conflict zones, especially in developing countries such as Sudan [2], Ethiopia [3,4], Nigeria [5], and currently in the Gaza Strip [2]. Factors such as restricted access to food, destruction of agricultural resources, displacement, and disrupted markets exacerbate these issues, leading to widespread malnutrition [3–6]. The recent war on Gaza further intensified this crisis, with an estimated 1.8 million people facing extreme hunger [6,7]. Prior to this conflict, food insecurity was already prevalent in Gaza, particularly among families with young children [8]. This chronic food insecurity, characterized by inadequate quantity and quality of food, may have long-term consequences, including epigenetic effects on the health of current and future generations [9].

The interplay between ongoing political conflict and food insecurity necessitates a comprehensive assessment to understand the full extent and implications of hunger in the region [7]. The period following October 7th has seen numerous reports of severe shortages in food availability, restricted access to essential goods, and a significant rise in the number of individuals and families experiencing hunger [10,8]. These developments underline the urgency for empirical research to quantify and analyze the prevalence of hunger and famine in Gaza, providing a basis for targeted interventions and policy responses.

Exposure to war and crisis leads to hunger, food insecurity, and famine [11]. Hunger is a personal, physical discomfort caused by a lack of food, while food insecurity is a broader issue of inconsistent access to nutritious food. Famine, the most severe form of food insecurity, is a catastrophic shortage of food that causes starvation and death. Addressing food insecurity early is crucial to prevent escalation to famine, which results in severe food shortages, high death rates, and humanitarian crises requiring urgent aid.

The Household Food Security Survey Module (HFSSM) is also a pivotal yet contested tool for assessing food security and hunger. It is a comprehensive tool designed to assess household food security by evaluating the frequency and severity of food access-related difficulties. It consists of a series of questions that capture experiences of inadequate food access, including concerns about food availability, the need to reduce meal sizes, and reliance on less desirable food options. The HFSSM plays a crucial role in identifying food insecurity levels within households, providing critical data that can inform policy decisions and program development for addressing food insecurity. By highlighting the specific challenges faced by families, HFSSM enables targeted interventions to improve food security and the overall well-being of affected populations [9].

The Household Food Insecurity Access Scale (HFIAS) is a tool used to measure the level of food insecurity experienced by households. It assesses the accessibility and availability of food by gauging the frequency and severity of food-related problems, such as worry about food shortages and the inability to eat preferred foods. By identifying the degree of food insecurity, HFIAS provides valuable insights for policymakers and humanitarian organizations, informing targeted interventions and resource allocation. Understanding food insecurity through HFIAS can lead to improved food security strategies and enhanced support for vulnerable populations, ultimately contributing to better health and well-being outcomes [12].

The Household Hunger Scale (HHS) is a significant tool in assessing the prevalence of famine, hunger, and food insecurity; it measures the degree of hunger within a household based on a set of standardized questions, providing a direct assessment of hunger by focusing on experiences and behaviors associated with food scarcity [13]. The HHS identifies and categorizes the severity of hunger, distinguishing between mild, moderate, and severe hunger, thus offering nuanced insights into food insecurity [11]. In an era of extreme scarcity of humanitarian funding, HHS has substantial implications for resource allocation and humanitarian prioritization [11].

The findings of this research are intended to provide policymakers and humanitarian organizations with actionable insights, such as identifying the most vulnerable populations, quantifying the impact of specific restrictions, and evaluating the effectiveness of existing aid programs, to inform the development and implementation of targeted interventions that improve long-term food security and alleviate human suffering in Gaza. Further, the findings of the current work will inform policymakers, humanitarian organizations, and international bodies, facilitating the development of effective strategies to mitigate hunger and enhance food security in the occupied Strip amidst the ongoing unequalized, damaging war. Through this assessment, we aim to highlight the urgent need for coordinated efforts to address the humanitarian crisis, support the resilience and recovery of the affected communities in the Gaza Strip, and, most importantly, address the root causes of the conflict.

## Methods

### Study settings, population, and tools

A cross-sectional design was followed in assessing the prevalence of food insecurity and hunger in the Gaza Strip from May to July 2024, after seven to nine months of the Israeli military attacks in response to the 7th October attack by Hamas.

The study applied quantitative research methodology, using a non-probability convenience sampling technique, covering the Palestinian households in the five governorates of the Gaza Strip that are the northern governorates (North Gaza, Gaza City), the middle (Deir Al Balah), and the southern (Khan Younis, Rafah).

Data was collected through face-to-face interviews with household members using a paper-based questionnaire administered by experienced data collectors recruited from senior nutrition students within the Clinical Nutrition and Dietetics Department at Al-Azhar University-Gaza. These students underwent a rigorous training program to ensure data quality and consistency. The training encompassed a comprehensive review of the questionnaire, covering all sections, questions, and response options. Standardized data collection procedures were meticulously explained, including interview techniques such as active listening, probing, and clarification techniques to ensure accurate and complete data collection, as well as strategies for probing sensitive information ethically. Guidelines for accurate and timely data entry and management were also provided. Ethical principles, including informed consent, confidentiality, and participant privacy, were thoroughly discussed. A pilot test of the data collection process was conducted with a small sample of participants, and feedback from the pilot test was analyzed to make necessary adjustments to the questionnaire and data collection procedures. This intensive training program equipped the data collectors with the required knowledge and skills to conduct high-quality data collection, minimizing potential biases and ensuring the reliability and validity of the study findings.

The Palestinian Ministry of Health in the Gaza Strip granted ethical approval. After discussing the study's aims, advantages, risks, information confidentiality, and voluntary nature of participation, informed consent was obtained from all participants. Furthermore, all data collectors and investigators ensured the confidentiality of the information gathered from each study participant by using code numbers instead of personal identifiers and making the questionnaire inaccessible to anyone other than the investigators.

The inclusion criteria for this study included any citizen residing in the Gaza Strip for at least six months prior to the onset of the study, ensuring familiarity with local food systems and coping mechanisms. To directly capture the impact of the recent Israeli war, participants must have resided in Gaza throughout the conflict period, beginning on October 7th, ensuring their experiences directly reflect the food security challenges arising from the war's impact. Participants must express clear willingness to share detailed information about their food access experiences during the conflict, accurately report on their household's hunger levels and any changes experienced, describe coping strategies employed to address food scarcity, and provide informed consent to participate in the study, understanding the purpose, procedures, and potential risks involved.

For sample size calculation, we used the single population proportion formula by using Epi Info StatCalc considering the following assumptions: 95% confidence level (Zα/2), 32.4% proportion of households fell into the household hunger categories [14] for prevalence (p) and 5% margin of error (d), which was 337. With a 30% non-response rate and a three-size effect, the calculated sample size was 1213 households. A non-probability convenience sampling technique was used to select the required sample of households from all the affected parts of the Gaza Strip.

For this study, experts in both English and Arabic were involved in the translation process of the three assessment tools (HFSSM, HFIAS, HHS). The initial step involved translating the original English versions of the questionnaires into Arabic. To ensure the accuracy and reliability of the translated versions, a rigorous back-translation process was implemented. Independent translators subsequently translated the Arabic versions back into English. A thorough comparison was then conducted between the original English versions and the back-translated English versions to identify and address any discrepancies or inconsistencies. This meticulous approach aimed to ensure the semantic equivalence and cultural appropriateness of the Arabic versions of the questionnaires. A copy of the finalized Arabic questionnaire used in this study can be accessed through the repository link. To optimize resource utilization and minimize costs, the questionnaire was designed in a compact format, utilizing two condensed pages with two columns per page. This approach was necessary due to the scarcity of office printers, ink, and paper, as well as limited financial resources within the study context.

## Sociodemographic data

The sociodemographic section of the questionnaire included questions about the respondent's age, sex, and educational level, as well as that of their spouse. It also gathered information on marital status, the governorate of residence (before the war), household composition and the number of children under care, parental responsibilities, number of displacements experienced, self-reported socioeconomic status prior to the war, and the extent of house destruction. Additionally, the questionnaire included questions regarding the nutritional status of the siblings, specifically assessing signs of malnutrition such as weakness, noticeable weight loss, and inability to move. It also inquired about the number of affected children and whether any children under care had passed away due to lack of food and starvation.

The classification of socioeconomic status (Low, Medium, High) was based on self-reported assessments by participants regarding their socioeconomic situation before the war. To assist participants in their evaluation, they were given a brief description of each category: Low individuals or households experienced significant socioeconomic hardship and struggled to meet basic needs; Medium individuals or households had moderate socioeconomic stability and were able to meet basic needs but with limited resources for savings or discretionary spending; High individuals or households had substantial socioeconomic resources, able to meet basic needs comfortably and have significant discretionary income. The description emphasized key indicators such as income sources and levels, household assets, access to essential services, and typical consumption patterns. Participants were encouraged to consider their main sources of income (e.g., employment, business, agriculture) and their approximate income levels before the war, as well as their ownership of essential assets such as housing, vehicles, and livestock. Furthermore, participants were prompted to reflect on their prewar access to critical services like healthcare, education, and utilities, as well as their typical consumption patterns, such as their ability to afford a balanced diet, clothing, and other essential goods. By providing these guiding questions, the study aimed to ensure a more consistent and reliable self-reported assessment of socioeconomic status among participants, which directly impacts food security. This classification was derived and modified from the examined scales for the socioeconomic status [15].

For the displacement question, participants were asked about the number of displacements they experienced during the war, with displacement defined as the forced or involuntary movement of individuals from their usual residences to other, perceived safer locations due to the ongoing military actions and airstrikes by Israeli forces. These include internal displacement within Gaza or forced displacement due to military action. Data was collected on the frequency and duration of displacement episodes, as well as the type of displacement, including displacement to shelters, displacement to relatives' homes, or displacement to other locations. Displacement was hypothesized to significantly impact food security by disrupting access to food sources, markets, and livelihoods. This led to the loss of food stocks and increasing reliance on external food assistance. The study will investigate the relationship between displacement experiences and food insecurity outcomes, considering factors such as the frequency, duration, and type of displacement while controlling for other relevant factors such as age, gender, and pre-existing socioeconomic vulnerabilities.

## Food insecurity and hunger assessment and scoring systems

**The HFSSM.** The HFSSM tool consists of 18 questions; 11 of them are yes/no questions, and 7 of them have a 4-point Likert scale from never to often, where never and rarely are considered zero scores and sometimes given one mark score. The maximum total score was 18 points, and then it was categorized into two categories, with a score of 1 or more being enough to indicate food insecurity. So, only the participants who got a score of zero indicated food security [16,17].

**The HFIAS.** HFIAS tool consists of 9 questions using a 4-point Likert scale (never = 0, rarely = 1, sometimes = 2, often = 3). The maximum total score was 27 points, and this total score was categorized into four categories as follows: Food Secure, Mildly Food Insecure Access, Moderately Food Insecure Access, and Severely Food Insecure Access. Calculations used in identifying the four categories are found in the main reference [12].

**The HHS.** The HHS consists of three simple questions about the experience of extreme hunger with a yes/no answer, and if the answer is yes, there is a follow-up question about the frequency. The recall period is the previous 30 days. The HHS tool contains only three questions 4-point Likert scale (never = 0, rarely, and sometimes = 1, often = 2). The total possible score was 6 points. The cutoff points were as follows: 0–1; little or no household hunger; 2–3; moderate household hunger; 4–6; severe household hunger [13].

## Self-reported anthropometric measurements

Self-reported anthropometric measurements, comprising height and weight, were questioned before and during the war. These measurements were used to calculate and categorize the body mass index (BMI, kg/m²). BMI categories, based on the definition of the WHO, include underweight (BMI < 18.5), normal weight (BMI 18.5–24.9), overweight (BMI 25–29.9), and obesity (BMI ≥ 30 kg/m²), enabling further insight into participants' weight status [18]. The validity and reliability of self-reported anthropometric measurements are well documented in many published articles and have been systematically reviewed in one article by Fayyaz et al. [19].

To assess changes in self-reported body weight and calculated BMI before and after 7–9 months of the war on the Gaza Strip, a pairwise analysis was conducted. For each participant with complete data for both time points (pre-war and during-war), the change in body weight and BMI was calculated by subtracting the pre-war from the during-war value. The resulting paired differences in body weight and BMI were subsequently analyzed using paired t-tests to evaluate the statistical significance of the observed changes. Descriptive statistics and visualizations were used further to describe the magnitude and direction of these changes.

## Statistical analysis

Data were analyzed using Statistical Package for Social Sciences (SPSS) software, version 29 (IBM Corp, Armonk, NY, USA). Categorical variables were reported as frequencies and percentages, whereas continuous variables were described as mean ± standard deviation (SD). All of the scale questions were expressed as frequencies and percentages of the responses among participants. The total score was calculated as mentioned above in the scoring paragraph, and the cutoff points were applied to categorize the total score into its categories. The correlation between the total score of the HFSSM, HFIAS, and HHS scales and the sociodemographic data was determined by using the correlation test. Cross-tabulation and Chi-square analysis were used to investigate the categories of the HHS scale among sociodemographic variables. A logistic regression analysis determined significant predictors of suffering from hunger. A confidence interval (CI) of 95% was applied to represent the statistical significance of the results, and the level of significance was predetermined as $P < 0.05$.

## Results

One thousand five hundred-three households from the five governorates were surveyed. After data cleaning and removing participants with missing data, 1209 households were included in the final statistical analysis. These 1209 households were presented by 1209 representatives (aged 38 ± 9.6 years, 53.5% females). The final analysis excluded 294 households primarily due to missing or incomplete data, which were essential for accurate analysis. Some households could not provide the necessary information, while others faced logistical challenges, such as access issues caused by ongoing Israeli military attacks or displacement. Additionally, some respondents chose not to complete their participation due to safety concerns or mistrust of the survey process.

Table 1 summarizes the demographic and socioeconomic characteristics of participants, showing that the majority, aged 19–39, were females and predominantly from the Middle governorate. Most participants were married and mainly responsible for families headed by men, with a significant number experiencing total house destruction due to conflict. Households were primarily from the Middle region (Deir Al Balah), which accounts for 65.0%. The Northern region (North

 

**Table 1. Sociodemographic characteristics of the study participants and their represented households (n = 1209).**

| Variable | | Frequency | Percentage |
|---|---|---|---|
| **Age (**years) 37.97 ± 9.62 | 19-39 | 701 | 58.0 |
| | 40-59 | 508 | 42.0 |
| **Sex** | Male | 562 | 46.5 |
| | Female | 647 | 53.5 |
| **Governorate/City** | Northern (North Gaza and Gaza City) | 331 | 27.4 |
| | Middle (Deir Al Balah) | 786 | 65.0 |
| | Southern (Khan Younis and Rafah) | 92 | 7.6 |
| **Responsible for the family** Number of family members (5.63 ± 1.79) Number of dependent children (3.09 ± 1.59) | Man | 1087 | 89.9 |
| | Woman | 122 | 10.1 |
| **Self-reported socioeconomic status (before the war)** | Low | 412 | 34.1 |
| | Medium | 698 | 57.7 |
| | High | 99 | 8.2 |
| **Marital status** | Married | 1161 | 96.1 |
| | Widowed | 33 | 2.7 |
| | Divorced | 15 | 1.2 |
| **Partner** | | | |
| **Age of partner** (38.36 ± 10.28) (range 16–95 years) | Below than 19 | 12 | 1.0 |
| | 19-39 years | 674 | 55.7 |
| | 40-59 years | 484 | 40.1 |
| | 60 and above | 39 | 3.2 |
| **Partner working status** | Yes | 457 | 37.8 |
| | No | 752 | 62.2 |
| **The educational level of the partner** | Primary | 35 | 2.9 |
| | Elementary | 143 | 11.8 |
| | Secondary | 384 | 31.8 |
| | B.Sc. | 593 | 49.0 |
| | M.Sc. & Ph.D. | 54 | 4.5 |
| **House status and address** | | | |
| **House destruction status** | Total destruction | 657 | 54.3 |
| | Partial destruction | 372 | 30.8 |
| | No Destruction | 98 | 8.1 |
| | I do not know | 82 | 6.8 |
| **Address before the war** | City House | 981 | 81.1 |
| | Camp | 228 | 18.9 |
| **Current address** Number of displacements (4.45 ± 2.49) | Tent | 707 | 58.5 |
| | Home | 208 | 17.2 |
| | School | 295 | 24.3 |

Gaza and Gaza City) accounted for 27%, while the Southern region (Khan Younis and Rafah) represented about 8%. Most households were headed by men (about 90%), with an average of 5.63 ± 1.79 individuals per family household and 3.09 ± 1.59 dependent children. Before the war, 34% of the participants were of low socioeconomic status, 58% were of medium status, and about one-fifth were living in camps while the rest were living in city houses. A vast majority of the

participants were married (96.0%), with small percentages being widowed or divorced. Among partners (38.36 ± 10.28, range 16–95 years), 62.2% were not working, and the most common educational level was a Bachelor's degree (49.0%). The Israeli attacks caused complete house destruction for 54.3% of the households and partial destruction for 30.8%, with about 59% currently living in tents and 24% in schools. About 78% received intermittent help from relief organizations, 19.4% received no help, and 2.5% received regular help. During the seven to nine months of the war, the household families experienced 4.45 ± 2.49 displacements across the Strip regions (Table 1).

The demographics of our study reflect the overall population in Gaza, with a significant proportion of adults in their mid-to-late thirties and a fairly balanced sex ratio. Our analysis revealed notable disparities in food insecurity based on sex, with female-headed households experiencing higher levels of food insecurity than male-headed households, highlighting the unique challenges women face in accessing resources during the ongoing Israeli military attacks. Age-based differences were also evident, as participants under 30 reported higher food insecurity levels compared to older age groups, likely due to limited employment opportunities and increased reliance on family support. Overall, our findings indicate that demographics significantly influence food insecurity experiences in Gaza, emphasizing the importance of these interactions for informing targeted interventions and assistance strategies.

Table 2 reveals that a significant majority of children exhibited symptoms of starvation, with very minor reported deaths due to starvation. The data also indicates a notable weight loss among participants since the war, reflecting a shift in nutritional status, including increased underweight cases. A significant majority, about 84% (n = 1013) of the households' children exhibited one or more of the starvation symptoms (significant weight loss, fatigue, weakness, irritability, and a decreased immune response), with an average number of children who have symptoms 1.91 ± 1.47 per household. Very few children, 0.4% (n = 5) of households\ children passed away because of starvation, as reported by their parents, with average number of children who passed away because of starvation was 0.01 ± 0.17 per household.

In terms of assistance from relief organizations, about one-fifth (19.4%, n = 235) reported receiving no help, while the vast majority (78%, n = 944) received intermittent assistance. Overall, the data indicates a high prevalence of starvation symptoms among children. Most families received intermittent assistance from relief organizations, with a notable proportion not receiving any help. As depicted in Fig 1, before the war, the mean body weight was 74.8 ± 15.9 kg. On average, household individuals lost 10.5 ± 8.5 kilograms since the war started, resulting in a mean current self-reported body weight of 64.8 ± 15.2 kg. The mean current BMI was 22.8 ± 5.2 kg/m², while the mean before the war was 26.4 ± 5.4. In terms of BMI categories, the current distribution shows that about 16% of individuals were underweight, 58% had a normal weight,

Table 2. Malnutrition and anthropometric characteristics of the surveyed households (n = 1209).

| Variable | | Frequency | Percentage |
|---|---|---|---|
| **Starvation symptoms appearance** The number of children who have symptoms (1.91 ± 1.47) | Yes | 1013 | 83.8 |
| | No | 196 | 16.2 |
| **Did any child die because of starvation?** Number of children who die because of starvation (0.01 ± 0.17) | Yes | 5 | 0.4 |
| | No | 1204 | 99.6 |
| **Do you receive any help from a relief organization?** | No | 235 | 19.4 |
| | Regularly | 30 | 2.5 |
| | Intermittent | 944 | 78.1 |
| **BMI (kg/m²) categories [n(%)]** | | Current | Before war |
| Underweight (18.5–24.9) | | 188 (15.6) | 49 (4.1) |
| Normal (25.0–29.9) | | 698 (57.7) | 461 (38.1) |
| Overweight (30.0–34.9) | | 255 (21.1) | 447 (37.0) |
| Obese (>35.0) | | 68 (5.6) | 252 (20.8) |

21% were overweight, and 6% were obese. Before the war, these values were about 4%, 38%, 37%, and 21%, respectively. These results indicate a significant shift in the weight and BMI categories of the individuals, with a general trend of weight loss and a reduction in the proportion of overweight and obese individuals following the war (Table 2).

Table 3 outlines the experiences of households and children with food insecurity. Over half of respondents often worry about food running out and not being able to afford balanced meals. Most adults reported cutting meal sizes or skipping meals, while a significant percentage of children also faced meal reductions and hunger, with the vast majority reporting cutting meal sizes. The overall HFSSM score indicates that all participants experience food insecurity, with an average score of 14.54. The HFSSM survey questions about experiences with food insecurity in the past year. Most respondents (over 50%) reported worrying about running out of food before they had money to buy more, did not have money to get more food, and more than two-thirds were not able to afford balanced meals (Table 3). For questions specifically about adults, nearly all respondents (over 95%) in the households cut meal sizes or skipped meals at some point in the last year and, particularly, in three or more months, and reported that they ate less than they feel should do. About 80% of interviewed household adults experienced hunger but did not eat, and over 90% reported that they lost body weight. Less than half (43%) of respondent household adults reported that they did not eat for the whole day, while about one-third (32.5%) reported that they did not eat for an entire day in three or more months duration. For questions concerning children, the most common response (about two-thirds) was that children relied on a few cheap foods, and their parents could not feed them balanced meals. Nearly as many respondents (over 55%) said their children were not eating enough, and over 80% reported having to cut their children's meal sizes or described their children being hungry or having their children skip meals, especially during three or more months (about 80%). Half of the respondents said their children went a whole day without eating at some point in the last year. The survey concludes that 100% of the sample was food insecure (Table 3).

Table 4 reveals a critical food security situation within the surveyed population. A significant proportion of households experienced various forms of food insecurity, including frequent worries about food availability, limited food choices, and the need to reduce meal sizes or skip meals due to insufficient food. Worryingly, a high prevalence of hunger was observed, with a substantial number of households reporting going to sleep hungry and even going a whole day and night without eating. The results show that almost all (98%) of the sample was severely food insecure, with 2.3% experiencing moderate food insecurity. No respondents were categorized as mildly food insecure. The total HFIAS score was 19.85±4.54 (range 4.0–27.0), implying an extremely high level of food insecurity (Table 4).

(a)

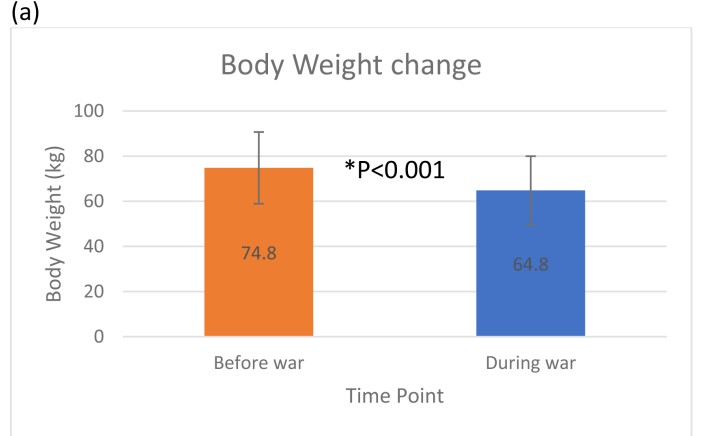

(b)

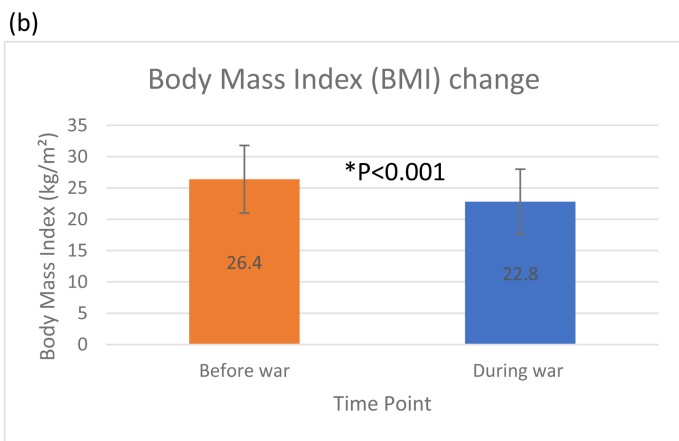

**Fig 1. Impact of the war on the Gaza Strip on the anthropometric measurements after 7-9 months.** (a) Mean change in self-reported body weight (kg) before and during war. (b) Mean change in the calculated body mass index (BMI, kg/m$^2$) before and during war. Error bars represent standard deviation (SD). *A significant difference (P <0.001, paired t-test) between before and during the war on the Gaza Strip.

**Table 3. HFSSM questionnaire for food insecurity (18 items).**

| Questions | | | Never | Rarely | Sometimes | Often |
|---|---|---|---|---|---|---|
| Household | Q1 | Worry that food would run out before (I/we) got money to buy more | 67 (5.5) | 82 (6.8) | 428 (35.4) | 632 (52.3) |
| Household | Q2 | The food bought did not last, and (I/we) did not have money to get more | 43 (3.6) | 138 (11.4) | 396 (32.8) | 632 (52.3) |
| Household | Q3 | Could not afford to eat balanced meals | 47 (3.9) | 55 (4.5) | 276 (22.8) | 831 (68.7) |
| | | | Yes | | No | |
| Adult | Q4 | Adult(s) cut the size of meals or skipped meals | 1175 (97.2) | | 34 (2.8) | |
| Adult | Q5 | Adult(s) cut the size of meals or skipped meals in three or more months | 1150 (95.1) | | 59 (4.9) | |
| Adult | Q6 | Respondents ate less than they felt they should | 1171 (96.9) | | 38 (3.1) | |
| Adult | Q7 | The respondent was hungry but did not eat | 944 (78.1) | | 265 (21.9) | |
| Adult | Q8 | Respondent lost weight | 1118 (92.5) | | 91 (7.5) | |
| Adult | Q9 | Adult(s) did not eat for the whole day | 523 (43.3) | | 686 (56.7) | |
| Adult | Q10 | Adult(s) did not eat for a whole day in three or more months | 393 (32.5) | | 816 (67.5) | |
| | | | Never | Rarely | Sometimes | Often |
| Child | Q11 | Relied on a few kinds of low-cost food to feed the child(ren) | 44 (3.6) | 66 (5.5) | 282 (23.3) | 817 (67.6) |
| Child | Q12 | Could not feed the child(ren) balanced meals | 25 (2.1) | 72 (6.0) | 315 (26.1) | 797 (65.9) |
| Child | Q13 | The child (ren) was not eating enough | 43 (3.6) | 105 (8.7 | 379 (31.3) | 682 (56.4) |
| | | | Yes | | No | |
| Child | Q14 | Cut size of child(ren's) meals | 1032 (85.4) | | 177 (14.6) | |
| Child | Q15 | Child(ren) were hungry | 1017 (84.1) | | 192 (15.9) | |
| Child | Q16 | Child(ren) skipped meals | 1033 (85.4) | | 176 (14.6) | |
| Child | Q17 | Child(ren) skipped meals in three or more months | 960 (79.4) | | 249 (20.6) | |
| Child | Q18 | The child (ren) did not eat for the whole day | 598 (49.5) | | 611 (50.5) | |

The total HFSSM score was 14.54±2.83 (range 3.0–18.0), and 100% of the sample size is considered to be food insecurity.

Table 5 summarizes the experiences of household hunger among respondents, indicating that a significant percentage often have no food available and go to sleep hungry. The low HHS score denotes that about half of the participants experienced severe household hunger, while less reported moderate hunger, with a small group experiencing little or no hunger. Over a third (33.7%) of respondents reported having no food to eat at all at some point in the last 30 days. Similarly, over a third (35.4%) said they went to sleep hungry, and nearly one-fifth (20%) used to go a whole day and night without eating. The survey results indicate an extremely high prevalence of hunger within the sample. While 11.7% reported little to no household hunger, the vast majority (89%) experienced hunger in varying degrees, with about 41% experiencing moderate hunger and nearly half (48%) experiencing severe hunger in their household during the last 30 days. Fig 2 shows the prevalence of hunger and food insecurity before and during the war. Data on the prevalence of hunger and food insecurity before they were derived from the UN Food and Nutrition Fact Sheet-Palestine [20] and the IPC on acute malnutrition in the Gaza Strip [6].

The correlation examines the relationship between three measures of hunger and food insecurity (HHS, HFSSM, HFIAS) and various sociodemographic determinants. There was a weak but statistically significant positive correlation between the number of times a household was displaced and all three hunger/food insecurity scores. This means that more displacement was associated with higher hunger and food insecurity scores. For the age factor, there were weak but statistically significant positive correlations between age and all three hunger/food insecurity scores. This suggests that older individuals may be more likely, but not necessarily, to experience hunger or food insecurity. The age of the partner also shows a similar correlation with HHS and HFIAS scores. For children's health, the number of children with symptoms shows a weak but statistically significant positive correlation with all three hunger/food insecurity scores, implying a link between children's health and household hunger and that the food security status of the household directly impacts children (S1 Table).

**Table 4. Responses to and a score of the HFIAS questionnaire (9 items).**

| Questions | | Never | Rarely | Sometimes | Often |
|---|---|---|---|---|---|
| Q1 | Worry that the household would not have enough food | 68 (5.6) | 76 (6.3) | 481 (39.8) | 584 (48.3) |
| Q2 | Not able to eat the kinds of food preferred | 42 (3.5) | 39 (3.2) | 174 (14.4) | 954 (78.9) |
| Q3 | Eat a limited variety of foods | 112 (9.3) | 61 (5.0) | 303 (25.1) | 733 (60.6) |
| Q4 | Eat some foods that you really did not want to eat | 30 (2.5) | 85 (7.0) | 285 (23.6) | 809 (66.9) |
| Q5 | Eat a smaller meal than you feel you need | 67 (5.5) | 100 (8.3) | 364 (30.1) | 678 (56.1) |
| Q6 | Eat fewer meals in a day | 48 (4.0) | 115 (9.5) | 299 (24.7) | 747 (61.8) |
| Q7 | No food to eat of any kind in your household | 174 (14.4) | 254 (21.0) | 373 (30.9) | 408 (33.7) |
| Q8 | Go to sleep at night hungry | 148 (12.2) | 247 (20.4) | 428 (35.4) | 386 (31.9) |
| Q9 | Go a whole day and night without eating | 298 (24.6) | 352 (29.1) | 313 (25.9) | 246 (20.3) |
| | | Frequency | Percent | | |
| **The total HFIAS score was 19.85±4.54 (range 4.0–27.0)** | | | | | |
| | Severely Food-Insecure Access | 1181 | 97.7 | | |
| | Moderately Food-Insecure Access | 28 | 2.3 | | |
| | Mildly Food-Insecure Access | 0 | 0.0 | | |
| | Food Secure | 0 | 0.0 | | |

The HFIAS tool consists of 9 questions, each on a 4-point Likert scale (never = 0, rarely = 1, sometimes = 2, often = 3).

The maximum total score was 27 points, and this total score is categorized into four categories as follows:

1 = Food Secure if [(Q1 = 0 or Q1 = 1) and Q2 = 0 and Q3 = 0 and Q4 = 0 and Q5 = 0 and Q6 = 0 and Q7 = 0 and Q8 = 0 and Q9 = 0]

2 = Mildly Food Insecure Access if [Q1 = 2 or Q1 = 3 or Q2 = 1 or Q2 = 2 or Q2 = 3 or Q3 = 1 or Q4 = 1) and Q5 = 0 and Q6 = 0 and Q7 = 0 and Q8 = 0 and Q9 = 0]

3 = Moderately Food Insecure Access if [(Q3 = 2 or Q3 = 3 or Q4 = 2 or Q4 = 3 or Q5 = 1 or Q5 = 2 or Q6 = 1 or Q6 = 2) and Q7 = 0 and Q8 = 0 and Q9 = 0].

4 = Severely Food Insecure Access if [Q5 = 3 or Q6 = 3 or Q7 = 1 or Q7 = 2 or Q7 = 3 or Q8 = 1 or Q8 = 2 or Q8 = 3 or Q9 = 1 or Q9 = 2 or Q9 = 3]

The number of children who passed away due to starvation has a weak positive correlation with the HFSSM score but not with the other two measures. Weight loss since the war started shows a weak but statistically significant negative correlation with the HFIAS score only. This implies that people who lost more weight tend to have higher scores on the HFIAS, indicating greater food insecurity and the lack of food as the main cause of weight loss, nothing else. Current weight and BMI were negatively and significantly correlated with the HHS only, indicating the impactful effect of the war on reducing body weight as a consequence of the lack of food security and widespread hunger. The information from S1 Table suggests that factors like displacement, age, and children's health may be important when considering hunger and food insecurity in this population.

The cross-tabulation and chi-square analysis of HSS against sociodemographic variables reveal significant findings. The seriousness of household hunger varies significantly with the city (P < 0.001), and the results showed significant differences in household hunger levels across the three regions. In the Northern region (Gaza City and North Gaza), about a quarter (25.5%) reported varying degrees of hunger, with 130 households experiencing moderate hunger and 178 households facing severe hunger (P < 0.001). To a lesser extent, the Middle region (Deir Al Balah) reported varying degrees of hunger (247 households, about 20%), with about 156 households reporting moderate hunger and 91 reported experiencing severe hunger. Conversely, in the Southern region (Khan Younis and Rafah), 42% (513 households) reported varying degrees of hunger, with 207 facing moderate hunger and a notable 306 households experiencing severe hunger. Overall, the findings suggest a significant variation in household hunger levels across different governorates, with the Southern region exhibiting the highest percentages of severe hunger. There was no significant difference in hunger severity between males and females, implying that the sex of the interviewed household members is not a determinant factor, and the Israeli attacks did not differentiate between the sensitive and non-sensitive, vulnerable and

**Table 5. HSS questionnaire (3 items).**

| Questions | | Never | Rarely | Sometimes | Often |
|---|---|---|---|---|---|
| Q1 | No food to eat of any kind in your household | 174 (14.4) | 254 (21.0) | 373 (30.9) | 408 (33.7) |
| Q2 | Go to sleep at night hungry | 148 (12.2) | 247 (20.4) | 428 (35.4) | 386 (31.9) |
| Q3 | Go a whole day and night without eating | 298 (24.6) | 352 (29.1) | 313 (25.9) | 246 (20.3) |
| | | Frequency | Percent | | |
| The total HHS score was 3.35±1.44 (range 0.0–6.0) | | | | | |
| | Little or no household hunger | 141 | 11.7 | | |
| | Moderate household hunger | 493 | 40.8 | | |
| | Severe household hunger | 575 | 47.6 | | |

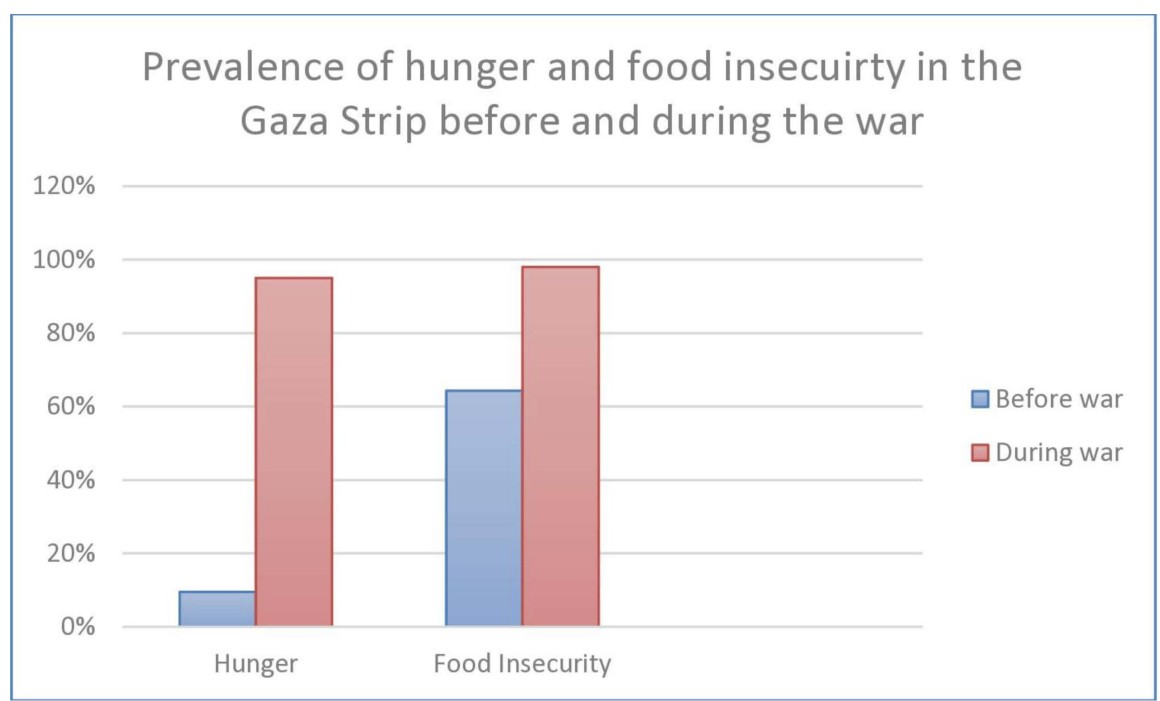

**Fig 2. Prevalence of hunger and food insecurity before and during the war.** Data on the prevalence of hunger and food insecurity before they were derived from the UN Food and Nutrition Fact Sheet-Palestine [20] and the IPC on acute malnutrition in the Gaza Strip [6].

non-vulnerable population groups. However, households where a woman was responsible for the family exhibited lower hunger severity (P<0.001).

Self-reported socioeconomic status before the war also played a crucial role (P<0.001), with households having medium socioeconomic status experiencing higher hunger severity. In contrast, expectedly, those with high financial status showed a lesser extent of hunger severity. The appearance of starvation symptoms was significantly associated with higher hunger severity (P<0.001). Hunger severity was higher in households where the partner was not working (P<0.001). The educational level of the partner was significantly related to hunger severity (P=0.003), with lower educational levels (primary and elementary) associated with higher hunger severity, while households with partners having BSc or higher degrees experienced less severe hunger. The condition of the house significantly affects hunger severity (P<0.001), with higher severity in households with total or partial house destruction (S2 Table). The address before the

war showed significant variation (P = 0.001), with higher hunger severity in households originally from city dwellers and less severe hunger in refugee camps. Current living conditions also play a significant role (P < 0.001), with higher hunger severity in households living in tents and less severe hunger in those living in homes. Lastly, receiving help from relief organizations significantly influences hunger severity (P < 0.001), with higher severity in households receiving intermittent help and less severe hunger in those not receiving or receiving regular help. Overall, the analysis demonstrates that household hunger severity is significantly impacted by various sociodemographic factors, including city, family responsibility, socioeconomic status, starvation symptoms, partner's working status, educational level, house condition, previous and current addresses, and relief organization support (S2 Table).

The logistic regression analysis of predictors of food hunger shows significant findings across various sociodemographic variables, which are presented in S3 Table. Using Northern Gaza (Gaza City and North Gaza) as the reference, the Middle region (Deir Al Balah) is about three times more likely to report hunger (OR, 2.771, 95% CI, 1.273 to 6.030, P = 0.010). The same, but to a lesser extent, for the Southern region (Khan Younis and Rafah) (OR, 1.930, 95% CI, 1.076 to 3.461, P = 0.027). These results indicate that individuals in Deir Al Balah and Khan Younis/Rafah were significantly more likely to experience food insecurity and hunger compared to those in Northern Gaza.

Sex is an important predictor, with females having lower odds of food hunger (OR = 0.073, 95% CI: 0.048–0.373, P = 0.011) compared to males (reference). Responsibility for the family significantly affects food hunger, with households where a woman is responsible having lower odds (OR = 0.144, 95% CI: 0.422–0.956, P < 0.001). Self-reported socioeconomic status before the war is also a significant predictor, with high financial status associated with higher odds of food hunger (OR = 2.989, 95% CI: 1.077–8.298, P = 0.036) compared to low socioeconomic status (reference). The appearance of starvation symptoms significantly predicts food hunger, with households not experiencing starvation symptoms having lower odds (OR = 0.187, 95% CI: -0.948 - -0.515, P < 0.001) (S3 Table).

The partner's working status is substantial, with households where the partner is not working having lower odds of food hunger (OR = 0.148, 95% CI: 0.273–0.605, P < 0.001). Address before the war is significant, with households from camps having higher odds of food hunger (OR = 4.167, 95% CI: 1.368–12.691, P = 0.012) compared to city residents (S3 Table). Overall, significant predictors of food hunger include the city of residence, sex, responsibility for the family, socioeconomic status before the war, the appearance of starvation symptoms, partner's working status, and address before the war. At the same time, other variables do not show significant predictive value. Graphical abstract summarizing the main changes in nutritional status, food insecurity, and hunger status before and during the active war in the Gaza Strip (Fig 3).

## Discussion

We are conducting this study in Gaza during Israeli military attacks to assess the rising food insecurity and famine resulting from the humanitarian crisis. This context enables us to gather timely data to inform humanitarian responses and policy interventions aimed at alleviating suffering and addressing urgent needs. Our research also aimed to provide actionable insights to shape both immediate and long-term support strategies for those affected by the conflict. Further, our study from May to July 2024, conducted after 7–9 months of Israeli military attacks, aimed to capture the evolving humanitarian crisis in Gaza. This timeframe allowed us to assess the immediate impact on food security, hunger, and famine, reflecting the acute challenges faced, such as supply disruptions, economic hardship, and displacement. Local and international responses also influenced the effectiveness of support for affected communities. We recognize that our findings represent a snapshot of a dynamic situation, emphasizing the need for ongoing research to monitor changes over time and inform immediate humanitarian efforts.

While this research was being conducted, no research was published on the topic to the authors' knowledge. It is also the first population-based study on the prevalence of hunger in the region following this damaging war on the Strip. By employing quantitative research, this paper seeks to provide a comprehensive understanding of the current food security

and hunger situation in the five governorates of the Gaza Strip, contributing to the broader discourse on humanitarian aid and conflict resolution.

Significant findings included high levels of food insecurity, with 100% of households experiencing some level of food insecurity according to HFSSM and 97.7% being severely food insecure per HFIAS. The results revealed a catastrophic, unprecedented, extremely high prevalence of hunger, with 95% of the surveyed people experiencing different forms of hunger. The HHS indicated that 88% of households experienced varying degrees of hunger. These astonishing numbers are comparable with the findings of relevant studies in other parts of the world, such as Nigeria and Ethiopia, where armed conflicts exist, such as Ethiopia, Nigeria, and Sudan [2–5,14,21].

The persistent food insecurity faced by Palestinian communities may reflect a broader pattern where access to essential resources is utilized as a means of exerting control, often aimed at breaking the will of the affected population. Reports and studies highlight how restrictions on movement, trade, and agricultural activities lead to increased hunger and malnutrition among those communities [22,23]. This tactic not only exacerbates humanitarian crises but also undermines the dignity and agency of individuals, further entrenching cycles of suffering and dependence. Ultimately, utilizing hunger as a weapon contravenes foundational human rights principles and calls for urgent international attention and action [24].

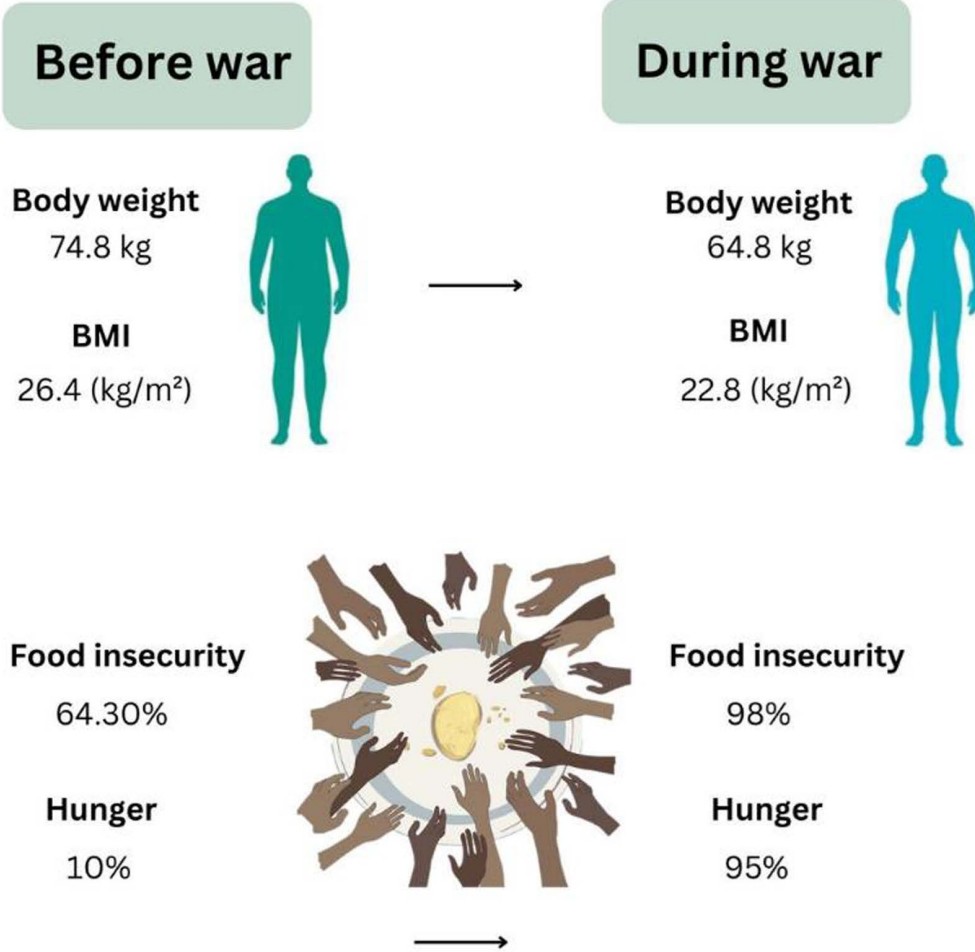

**Fig 3. Graphical abstract summarizing the main changes in nutritional status, food insecurity, and hunger status before and during the active war in the Gaza Strip.**

As per our results, the war has caused considerable weight loss, with the average weight decreasing from 74.6 kg before the conflict to 64.1 kg. While food insecurity is a primary factor, other key contributors include stress and trauma from ongoing attacks, which can alter eating habits and lead to anxiety and depression [25,26]. Limited access to healthcare exacerbates health issues, potentially causing further weight loss [27]. Additionally, the psychological impacts of living in a conflict zone can disrupt eating patterns, causing emotional eating or loss of appetite [28]. The chaotic environment also hinders regular food preparation and consumption. In summary, the interplay of food insecurity, stress, healthcare access, and psychological effects significantly contributes to the weight loss observed. This community-based cross-sectional study was conducted to assess household hunger during the Israeli war on Gaza 2023–2024. The study's target population was all the households in the Gaza Strip, Palestine. The Gaza Strip has long been a focal point of geopolitical conflicts and humanitarian crises. Since the mid-20th century, the region has been shaped by complex historical events, including the 1948 Arab-Israeli War, the 1967 Six-Day War, the subsequent Israeli occupation, and the recently aggressive damaging war on Gaza 2023–2024.

The current catastrophic hunger and famine in Gaza reflect both the ongoing war and the prolonged blockade that has lasted for 17 years since Hamas's election victory in 2006 [29]. This blockade has severely limited access to essential goods, including food, medical supplies, and building materials, exacerbating humanitarian crises and driving high poverty and unemployment rates. The local economy suffers from chronic shortages of basic goods, rampant poverty, and an almost non-existent manufacturing sector. In recent months, the situation in Gaza has deteriorated to an unprecedented level, culminating in a full-blown famine [30,31]. Several factors have contributed to this crisis, including intensified blockade restrictions on food and fuel imports, which have severely disrupted food supply chains. Agricultural production within Gaza has plummeted due to a lack of resources and infrastructure damage, further limiting local food availability [29,31,32].

Additionally, repeated aggressive wars have displaced nearly all families, destroyed homes, and obliterated agricultural lands, exacerbating food insecurity. Egypt's restrictions at the southern border further isolate the region. Humanitarian organizations have condemned the blockade as a form of collective punishment that violates international law, calling for a resolution that safeguards civilian rights while addressing security concerns [33]. The blockade, combined with recurrent violence, has devastated Gaza's infrastructure and economy, leaving the majority of its 2.3 million residents reliant on humanitarian aid, with youth unemployment exceeding 80% [34].

Our analysis showed that displacement, age, socioeconomic status, and education significantly impact hunger levels in distinct ways. Displacement was a key factor, with displaced individuals reporting higher food insecurity due to loss of livelihoods and support networks. Younger individuals, particularly those under 30, experienced greater food insecurity, likely due to limited employment opportunities. Socioeconomic status emerged as a major determinant, with lower-income households facing severe hunger amid the ongoing Israeli military attacks, which have disrupted job availability and income stability.

Interestingly, education had a complex relationship with hunger severity. While higher education typically correlates with better socioeconomic opportunities [35], individuals with some post-secondary education faced unique challenges in securing jobs during the conflict, leading to unexpected food insecurity levels. Our findings revealed that socioeconomic status significantly influenced hunger severity, more so than anticipated, highlighting the need for targeted economic interventions. These interconnected variables distinctly shape food insecurity experiences in Gaza, providing crucial insights for effective intervention strategies.

Our study enhances existing research on food insecurity in conflict zones such as Sudan, Ethiopia, and Nigeria [2–5], by providing updated data amid the current humanitarian crisis. While previous studies reported varying food insecurity levels [3,4,14,21], we observed significantly higher rates during our research period, which was attributed to several factors. The timing of our study aligns with a particularly intense phase of Israeli attacks, exacerbating vulnerabilities, increasing displacement, disrupting food supply chains, and heightening economic hardship for many conditions that earlier studies may not have fully captured.

Our research utilizes comprehensive methodologies and data collection techniques to understand better the current factors affecting food insecurity in Gaza, potentially leading to findings that contrast with prior studies [3,4,14,21]. Additionally, discrepancies may arise from differences in definitions and assessment tools used to evaluate food insecurity. By clarifying the current circumstances, our study highlights the severity of the crisis and informs more effective interventions. The unprecedented damage from Israeli attacks makes comparisons to past conflicts challenging, as similar conditions may not exist in contemporary history.

This unprecedented situation has been documented by Human Rights Watch, which asserts that (1) The Israeli government is employing starvation of civilians as a method of warfare in the Gaza Strip, constituting a war crime; (2) The Israeli officials have publicly stated their intention to deprive civilians in Gaza of essential resources such as food, water, and fuel, a goal that is evident in the military operations conducted by Israeli forces; and (3) The Israeli government must cease attacks on objects essential for the survival of the civilian population, lift its blockade of the Gaza Strip, and restore electricity and water services [24].

Several factors are believed to be associated with the devastating effect on food security and the high prevalence of hunger in the Gaza Strip due to the recent war. These factors include incessant shelling and ground operations that have inflicted significant damage or completely destroyed agricultural land and food production equipment, such as bakeries, mills, and food processing facilities, leading to the collapse of Gaza's food system [7]. The dire situation has forced several families, particularly in northern Gaza, to consume animal feed and plants to survive. Experts and human rights organizations widely regard Gaza as experiencing the most severe hunger crisis globally, with its population on the brink of famine [36], if not already experiencing it. Although international organizations are currently working to alleviate the severe food shortage and hunger, it is crucial to develop more comprehensive and lasting strategies to address the underlying causes of food insecurity in Gaza. This will help ensure that all inhabitants have access to a sufficient and nourishing diet [7].

In 2022, a report from the Gaza Strip revealed that children in food-insecure homes had a significant occurrence of moderate underweight (30.4%), stunting (32.8%), wasting (9.6%), and acute undernutrition (30.4%). This research was published approximately two years before October 7th. Before the damaging war, there were notable disparities in weight, height/length, mid-upper arm circumference, weight-for-age and mid-upper arm circumference z-scores, underweight, acute undernutrition, protein intake, fat intake, vitamin D intake, zinc intake, continued breastfeeding, nutrition-related knowledge, nutrition-related attitudes, and minimum dietary diversity score between the food-insecure and food-secure groups. In addition, approximately 56.0% of households experiencing food insecurity lack sufficient information about nutrition, 77.6% hold negative attitudes towards nutrition, and 95.2% fail to meet the minimum dietary diversity score. Overall, children from households experiencing food insecurity had a significant occurrence of moderate underweight, stunting, wasting, and acute undernutrition [7,37,34]. Furthermore, the combination of low economic status, inadequate dietary intake, insufficient knowledge, attitudes, and practices linked to nutrition, and a lack of variety in the diet all contributed to the elevated levels of food insecurity observed in children under the age of five in the Gaza Strip before the damaging war [37]. However, all these devastating conditions worsened after October 7th. Before the current situation, more than 75% of the Gaza Strip's population relied on assistance, as shown in the Global Nutrition Cluster Report of February 2024 [33], with reliance on humanitarian aid ranging from 70% to as high as 85% in the different governorates of the Strip. In the GNC report, food insecurity ranged from 60.9% in the Gaza governorate to 69.5% in the Rafah governorate. Food insecurity went in line with the assistance level in the five governorates of the Strip, where the highest rate of reliance on humanitarian assistance was in Rafah (85.1%), down to the lowest rate in Gaza (70.9%) [33].

Assessing the nutritional status of drivers in the Gaza Strip by the GNC report seven months ago showed a devastating situation that is consistent with the catastrophic current situation seen now [33]. The analysis of the different drivers for the nutritional status was as follows: Dietary Diversity in children 6–23 months and Dietary diversity in pregnant and breast-feeding women (PBW) were extremely critical in four out of the five main governorates (North Gaza, Gaza City, Deir Al

Balah, Khan Younis, and Rafah); Children reporting one or more diseases, Diarrhea (for children under five years, CU5), and other diseases (fever, vomiting, skin infection) were extremely critical in two governorates (Deir Al Balah and Rafah); and finally, water and sanitation access was extremely critical in three governorates (North Gaza, Deir Al Balah, and Rafah). However, acute respiratory infection in CU5 was crucial to Deir Al Balah and Rafah. These data reveal that Rafah and Deir Al Balah were the governorates most affected by malnutrition in the Gaza Strip [33].

Contradictions arise when Southern Gaza reports higher hunger rates than Northern Gaza, despite earlier assessments indicating more severe conditions in the north. This discrepancy can be understood through the geographical context and the original locations of displaced individuals. Historically, residents of Northern Gaza, especially in cities like Gaza City, have had a higher socioeconomic status and better access to income. This economic advantage has enabled many to retain savings, gold, or financial assistance, enhancing their ability to cope with crises and resulting in lower hunger prevalence. In contrast, individuals in Southern Gaza often lack similar resources, making them more susceptible to food insecurity. Additionally, many displaced people from Northern Gaza relocated to the Middle and Central areas during our data collection, further influencing hunger dynamics. These socioeconomic factors are crucial in understanding the varying hunger rates across different regions of Gaza.

A new report by the Integrated Food Security Phase Classification (IPC) confirms the catastrophic state of food insecurity in Gaza [10]. According to the report, the entire Gaza Strip remains at a high risk of famine due to ongoing war on the occupied Strip and restricted humanitarian access. The report indicates that 96% of the population, or 2.15 million people, are experiencing acute food insecurity (IPC Phase 3 or higher), with 495,000 individuals facing catastrophic levels of food insecurity (IPC Phase 5) until at least September 2024. The severity of the situation underscores the urgent need to ensure that food and other supplies reach all residents in Gaza. The report emphasizes that only a cessation of hostilities combined with sustained humanitarian access can mitigate the risk of famine in the Gaza Strip [10].

Based on our findings, we propose specific recommendations, including long-term strategies and quick-win interventions. First, targeted food assistance programs should be established to address the immediate needs of vulnerable populations, particularly displaced families and low-income households. Cash or food vouchers can empower families to choose based on their dietary needs and stimulate local markets. Second, nutrition education programs should be implemented to help families make informed choices about food preparation and healthy eating during periods of limited food access. For long-term strategies, invest in livelihood support initiatives to create job opportunities for younger individuals and women through training programs or small business support aligned with local market needs. Improving coordination among humanitarian organizations is vital for streamlining efforts and enhancing intervention effectiveness. Quick-win interventions can include community food distribution initiatives targeting neighborhoods most affected by food insecurity and community gardening projects to promote food sovereignty. By prioritizing these recommendations, humanitarian organizations can significantly improve food security and mitigate hunger in Gaza. Ultimately, ending the Israeli occupation and ensuring Palestinians' rights to live with dignity and freedom are essential for resolving these issues.

Beyond immediate relief efforts, several structural changes are essential for sustainable solutions. First, lifting the blockade on Gaza is crucial for enabling the free movement of people and goods, allowing farmers and businesses to access markets, and enhancing economic opportunities and food availability. Policies that promote economic development and invest in local agriculture can help create a self-sufficient food system, reducing reliance on external aid. Second, rebuilding essential services damaged by Israeli attacks, such as healthcare, education, and infrastructure, is vital for addressing the social determinants of food insecurity and boosting community resilience. Establishing a political dialogue aimed at achieving lasting peace and ending the Israeli occupation is also essential, recognizing Palestinian rights to live with dignity and freedom.

Our study underscores the urgent need for these policy changes. It highlights the inadequacy of current international humanitarian responses, which do not align well with the Sustainable Development Goals (SDGs), particularly Goal 2 (Zero Hunger) and Goal 1 (No Poverty). While humanitarian efforts provide crucial immediate relief, they fall short of

addressing systemic issues stemming from ongoing conflict and socioeconomic instability. Integrating humanitarian action with political solutions is necessary to ensure that international frameworks and rights extend beyond mere formal declarations. Without addressing these deeper structural issues, the objectives of the SDGs will remain unmet, and communities in Gaza will continue to suffer from severe food insecurity and human rights violations. By advocating for these essential policy changes and identifying gaps in current humanitarian approaches, our study aims to inform more effective and sustainable interventions aligned with international human rights and development frameworks.

Building upon Hassoun's recent analysis, which highlights severe environmental degradation, economic crippling, and social disruption, our research further underscores the catastrophic impact of the current conflict on the food security of the Gazan population. The blockade and ongoing hostilities have exacerbated existing vulnerabilities, leading to unprecedented levels of food insecurity, widespread hunger, and a heightened risk of famine. This aligns with Hassoun's observations, as the destruction of infrastructure, the crippling of the economy, and the disruption of social systems have all contributed to the erosion of food security and the overall well-being of the Gazan population [38].

Disseminating the study findings through channels such as social media, newspapers, and media outlets will effectively communicate the results to stakeholders, decision-makers, and policymakers, thereby raising awareness of food insecurity and hunger in Gaza. Well-crafted messages on social media, informative articles in newspapers, and impactful narratives in the media can broaden the audience's reach, including potential donors and supporters. Further, engaging policymakers with clear, data-driven insights can help shape policies that address the root causes of hunger, ensuring solutions are both immediate and sustainable. A robust dissemination strategy will significantly inform humanitarian organizations' actions and foster collaborative efforts to combat hunger effectively.

Though the current work presents a novel, first-of-its-kind original research work since the beginning of the war in the Gaza Strip after October 7th and covers a large sample size for the households in the occupied Strip, the current work entails a list of limitations that should be considered. While HFIAS offers a focused examination of food access and the severity of food insecurity through nine specific questions, its critics argue that it may oversimplify complex food insecurity experiences, especially in diverse cultural contexts [39]. On the other hand, HFSSM, with its comprehensive 18-question framework, provides a broader, more detailed picture of food security over the past year. However, detractors claim that its extensive nature can lead to respondent fatigue and potential inaccuracies [40]. Despite their widespread use in research and policy-making, both tools face scrutiny over their effectiveness and adaptability in different settings. This highlights the ongoing debate over the best methods to capture the multifaceted nature of food insecurity and hunger. Further, the ongoing conflict in Gaza presented several significant challenges and limitations in our data collection efforts. Firstly, the security situation greatly impacted logistics, making it difficult for our research team to access certain areas, particularly those experiencing heightened violence or instability. This often resulted in delays and necessitated adjustments to our planned survey locations. Secondly, participants' willingness to engage was also affected by the prevailing conditions. Many individuals expressed reluctance to participate due to security concerns, trauma from recent events, or a lack of trust in unknown researchers, even if they are from Gaza. This led to instances where potential respondents declined to be interviewed, which could introduce bias in our sample. Additionally, the disruptions in daily life caused by the destructive Israeli attacks, including displacement and uncertainty, impacted the availability of households for interviews, further complicating our data collection process. Despite these challenges, our team employed various strategies to ensure that we could conduct the study safely and ethically, including close coordination with influencing people (camp leaders and coordinators such as *Imams* or religious and tribal leaders) and adapting our methods to accommodate participants' needs and concerns.

While we collected self-reported anthropometric data, including weight changes, we acknowledge that stress and trauma from ongoing Israeli military attacks may affect reporting accuracy [28]. Participants may struggle to provide precise information or recall changes due to emotional impacts, which should be considered when interpreting our findings. Additional limitations include challenges in accessing participants due to security concerns, logistical difficulties,

and potential selection bias from displacement and restricted movement. Disrupted infrastructure and communication in conflict zones may hinder accurate and timely data collection. Recall bias is also a concern, as stress may affect participants' abilities to remember their food insecurity experiences accurately. The rapidly evolving situation in Gaza can lead to fluctuating food insecurity levels, complicating consistent data capture. Political and ethical dilemmas regarding sensitive data reporting may also influence findings, while limited funding could restrict the study's scope.

The study limitations also include potential biases associated with self-reported data. Social desirability bias may have influenced participants to provide answers that they perceived as socially acceptable or expected by the researchers, potentially leading to underreporting of negative experiences such as severe hunger or coping mechanisms that might be perceived as stigmatizing. Furthermore, reporting bias might have influenced participants to selectively report information that supports their perspectives or experiences, potentially leading to an overestimation or underestimation of certain aspects of food insecurity. Finally, the phrasing of questions or the interpretation of responses might have been influenced by cultural norms and sensitivities within the Gazan context, potentially leading to misinterpretations or inaccuracies in the data.

## Conclusions

The analysis of sociodemographic variables and their association with household hunger severity and food insecurity reveals a critical situation in Gaza, with unprecedentedly high levels of food insecurity and hunger approaching famine. Key predictors of hunger include city of residence, sex, family responsibilities, pre-war socioeconomic status, symptoms of starvation, partner's employment status, and pre-war address. Specifically, residents of Middle Gaza and camp dwellers are severely affected. Households headed by women, those with low socioeconomic status, and those showing starvation symptoms are particularly vulnerable. These findings underscore the dire consequences of Israeli attacks on Gaza, leading to infrastructure destruction and displacement. The reported high rates of household hunger severity and the urgent predictors of food insecurity highlight the need for urgent humanitarian intervention. Our recommendations target vulnerable populations, including low-income families, pregnant and lactating women, the elderly, and children. We aim to inform local NGOs, international humanitarian agencies, and policymakers to develop tailored strategies addressing the root causes of the catastrophic hunger and famine in the Gaza Strip. Further, immediate action is imperative, including a ceasefire and comprehensive relief efforts to meet the affected population's needs and rebuild Gaza's economic fabric. Decisive action is needed to alleviate the food insecurity crisis through sustained humanitarian assistance.

## Supporting information

**S1 Table. Correlation between the three food security/hunger assessment tools and the sociodemographic characteristics of the study households in the Gaza Strip.**
(DOCX)

**S2 Table. HHS vs. sociodemographic cross-tabulation and chi-square analysis.**
(DOCX)

**S3 Table. Predictors of food insecurity and hunger.**
(DOCX)

## Acknowledgments

The authors want to express our gratitude to all of the study participants and the Palestinian Ministry of Health for facilitating the study's execution. Thanks are due to Ms. Dania AlKawamleh for her assistance in drawing the figures and the graphical abstract.

## Author contributions

**Conceptualization:** MoezAlIslam Faris, Ayman S. Abutair.

**Data curation:** Ayman S. Abutair, Reham M. Elfarra, Amal M. Firwana, Rawan M. Firwana, Madleen M. AbuHajjaj, Shaimaa A. Shamaly, Samar S. AbuSamra, Hanan S. Bashir, Noor A. Abedalrahim, Noor A. Nofal, Mhran K. Alshawaf, Rania M. Al Shatali, Kafa I. Ghaben, Moayad I. Alron, Sara S. Alqeeq, Aya O. Al-Nabahin, Reem A. Badawi.

**Formal analysis:** Ayman S. Abutair.

**Funding acquisition:** MoezAlIslam Faris, Nida. A. Barqawi.

**Investigation:** MoezAlIslam Faris.

**Methodology:** MoezAlIslam Faris.

**Project administration:** MoezAlIslam Faris.

**Resources:** MoezAlIslam Faris.

**Supervision:** MoezAlIslam Faris, Ayman S. Abutair.

**Validation:** MoezAlIslam Faris.

**Writing – original draft:** MoezAlIslam Faris.

**Writing – review & editing:** Ayman S. Abutair.

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
