## [Decision Letter · Decision Letter 0]

19 Nov 2024

PONE-D-24-35303Catastrophic Hunger in Gaza: Unprecedented Levels of Hunger Post-October 7th. A Real Population-Based Study from the Gaza StripPLOS ONE

Dear Dr. Faris,

Thank you for submitting your manuscript to PLOS ONE. After careful consideration, we feel that it has merit but does not fully meet PLOS ONE’s publication criteria as it currently stands. Therefore, we invite you to submit a revised version of the manuscript that addresses the points raised during the review process.

We have evaluated your manuscript ID D-24-35303 entitled "Catastrophic Hunger in Gaza: Unprecedented Levels of Hunger Post-October 7th. A Real Population-Based Study from the Gaza Strip" which you submitted to PLOS ONE and require revisions before your article can be published. As you can see below, the reviewers provide comments. Please consider addressing these comments as an essential next step to take this manuscript forward.

We look forward to receiving your revised manuscript.

Kind regards,

Marianne Clemence, Staff Editor, on behalf of,

Dr. Mohammed Alkhaldi

Academic Editor

PLOS ONE

Journal Requirements: When submitting your revision, we need you to address these additional requirements. 1. Please ensure that your manuscript meets PLOS ONE's style requirements, including those for file naming. The PLOS ONE style templates can be found at https://journals.plos.org/plosone/s/file?id=wjVg/PLOSOne_formatting_sample_main_body.pdf and https://journals.plos.org/plosone/s/file?id=ba62/PLOSOne_formatting_sample_title_authors_affiliations.pdf 2. We note that your Data Availability Statement is currently as follows: All relevant data are within the manuscript and its Supporting Information files. Please confirm at this time whether or not your submission contains all raw data required to replicate the results of your study. Authors must share the “minimal data set” for their submission. PLOS defines the minimal data set to consist of the data required to replicate all study findings reported in the article, as well as related metadata and methods (https://journals.plos.org/plosone/s/data-availability#loc-minimal-data-set-definition). For example, authors should submit the following data: - The values behind the means, standard deviations and other measures reported;- The values used to build graphs;- The points extracted from images for analysis. Authors do not need to submit their entire data set if only a portion of the data was used in the reported study. If your submission does not contain these data, please either upload them as Supporting Information files or deposit them to a stable, public repository and provide us with the relevant URLs, DOIs, or accession numbers. For a list of recommended repositories, please see https://journals.plos.org/plosone/s/recommended-repositories. If there are ethical or legal restrictions on sharing a de-identified data set, please explain them in detail (e.g., data contain potentially sensitive information, data are owned by a third-party organization, etc.) and who has imposed them (e.g., an ethics committee). Please also provide contact information for a data access committee, ethics committee, or other institutional body to which data requests may be sent. If data are owned by a third party, please indicate how others may request data access. 3. Please include your full ethics statement in the ‘Methods’ section of your manuscript file. In your statement, please include the full name of the IRB or ethics committee who approved or waived your study, as well as whether or not you obtained informed written or verbal consent. If consent was waived for your study, please include this information in your statement as well. 4. Please include captions for your Supporting Information files at the end of your manuscript, and update any in-text citations to match accordingly. Please see our Supporting Information guidelines for more information: http://journals.plos.org/plosone/s/supporting-information. 5. Please review your reference list to ensure that it is complete and correct. If you have cited papers that have been retracted, please include the rationale for doing so in the manuscript text, or remove these references and replace them with relevant current references. Any changes to the reference list should be mentioned in the rebuttal letter that accompanies your revised manuscript. If you need to cite a retracted article, indicate the article’s retracted status in the References list and also include a citation and full reference for the retraction notice.

**Additional Editor Comments:**

Thank you for submitting your manuscript to PLOS ONE and I look forward to receiving your revision.

Dr. Mohammed Alkhaldi

**Comments from the Journal Office**

Please ensure that your Discussion is focused on the findings of your study rather than more general political discussion.

Reviewers' comments:

Reviewer's Responses to Questions

**Comments to the Author**

1. Is the manuscript technically sound, and do the data support the conclusions?

Reviewer #1: Yes

Reviewer #2: Yes

Reviewer #3: Partly

Reviewer #4: Yes

2. Has the statistical analysis been performed appropriately and rigorously? 

Reviewer #1: I Don't Know

Reviewer #2: Yes

Reviewer #3: I Don't Know

Reviewer #4: Yes

3. Have the authors made all data underlying the findings in their manuscript fully available?

Reviewer #1: Yes

Reviewer #2: Yes

Reviewer #3: No

Reviewer #4: Yes

4. Is the manuscript presented in an intelligible fashion and written in standard English?

Reviewer #1: Yes

Reviewer #2: Yes

Reviewer #3: Yes

Reviewer #4: Yes

5. Review Comments to the Author

Reviewer #1: This paper is an important contribution in documenting Israel's genocide against Palestinians. While I appreciate the authors' attempts to use "neutral" language to describe and document the levels of food insecurity and hunger currently experienced by Palestinians, it is critical that the authors consider the connections between the hunger and famine being experienced in Gaza, and the concept of starvation as a weapon of a genocidal war being perpetrated by Israel.

I would therefore like to recommend that the authors consider the report produced by Human Rights Watch on this matter: https://www.hrw.org/news/2023/12/18/israel-starvation-used-weapon-war-gaza. The report asserts that: (1) The Israeli government is using starvation of civilians as a method of warfare in the Gaza Strip, which is a war crime; and (2) Israeli officials have made public statements expressing their aim to deprive civilians in Gaza of food, water, and fuel – statements reflected in Israeli forces’ military operations.

On a personal note, as an author who has struggled to publish papers in peer reviewed journals because reviewers tend to question the neutrality and objectivity in my analysis of Israel's genocidal tactics, I completely understand your decision and approach with this paper. Consequently, my comments above should be considered as a recommendation rather than as a prerequisite for publication.

Reviewer #2: The manuscript is well written, the research study is of good quality, the topic is of high importance covering a pressing issue and it is necessary to have publications in this area. The access of researchers to Gaza is limited making the study very valuable.

Some minor writing mistakes need to be corrected

Page 10 I suggest to add some information on the history of the blockade on Gaza

Page 12 the repetition of the word report is redundant in this sentence "the report of the Global Nutrition Cluster (GNC) Report"

Page 12 GNS should be GNC

Page 15 Middle GA should be clarified

Reviewer #3: 1) General Comments:

This paper covers a highly relevant and urgent issue by examining hunger levels among Gaza residents after the October 7th military escalation. The study aims to provide population-based data that could be valuable for informing policymakers and humanitarian organizations on addressing hunger and food insecurity.

2) Detailed Comments:

Title:

The title contains redundancy in the word "Hunger." A clearer, more concise alternative is:

Catastrophic Hunger in Gaza: A Population-Based Study of Unprecedented Food Insecurity Post-October 7th.

Abstract:

• Background: The start is strong, but the ending mentions informing policymakers about strategies to mitigate hunger, which is not discussed in the paper. The purpose should instead emphasize providing insights into hunger levels and food insecurity rather than strategies.

• Methods: Sociodemographic data should be included in this section for completeness.

Introduction:

• The background of the Gaza crisis is well covered, but the section should include the most recent reports on hunger and malnutrition in Gaza.

• Definitions of key terms like hunger, famine, starvation, and food security should be included to distinguish these closely related concepts.

• The introduction goes into detail about the Household Hunger Scale (HHS) but merely mentions the Household Food Insecurity Access Scale (HFIAS) and the Household Food Security Survey Module (HFSSM). A more balanced coverage is needed.

• It’s important to clarify that the study's purpose is to inform policymakers rather than to create strategies, aligning with the abstract.

Materials and Methods:

• This is a cross-sectional study conducted in the Gaza Strip from May to July 2024, during the ongoing war in the region.

• There is no clear description of how data were collected or whether the data collectors were trained.

• The method of data collection (e.g., paper-based, Google Docs) should be specified. It's unclear how the questionnaire was administered (online or in-person), though later in the results section it is mentioned that households were interviewed.

• Participant eligibility criteria, informed consent processes, and an explanation of household hunger categories are absent.

• The HFIAS tool explanation could be made easier to understand through a table or appendix.

• The full questionnaire should be included in the appendix, and details about its development, validation, and scope (e.g., anthropometrics, sociodemographics) should be clearly outlined. It wasn’t clear how the questionnaire was built until we discovered that it was a combination of the 3 scales along with other questions.

• More details are needed on how anthropometric data were collected and organized.

• The translation of assessment tools should mention who performed the translations and their qualifications.

• After viewing the data, a major flaw is the inconsistency in the data categories and geographical or current address names, which should be standardized to allow for proper analysis.

Results:

• Some data, such as economic status, were not mentioned in the methods section but appear here for the first time. How was economic status categorized? What are the boundaries, and according to what?

• The presentation of results would benefit from visual aids like graphs or figures, particularly for illustrating shifts in BMI and weight loss.

• The conclusion contradicts the results by stating that camp dwellers have higher hunger severity, which is not reflected in the results.

• When discussing the governorates, it is unclear whether the household location refers to where they are currently due to continuous displacement or where they originally came from.

• The phrase "with particular attention toward the three or more months by about one-third of the respondents" is unclear. How was this determined? Was it part of the questionnaire or a note from interviewers?

• Results should be summarized clearly before diving into further detail.

Discussion:

• The discussion is too lengthy and redundant in parts, especially when detailing the situation in Gaza before the war, which is not particularly related to the research aim. It should focus more on the study’s findings about hunger and food insecurity, backed by current evidence.

• The omission of study limitations, such as challenges faced during data collection in a conflict zone (e.g., safety, displacement, internet access, movement), especially in Northern Gaza, is significant. For example, the Rafah invasion and evacuation occurred at the beginning of the research period. How did this affect the research?

• More references are needed to strengthen some statements, and comparisons to similar studies would enhance the discussion.

• Non-participation challenges due to internet and war-related issues, such as the inability to reach people in tents either in the field or online, should be highlighted.

• The discussion should emphasize how the study’s results can inform humanitarian organizations in addressing hunger.

• Contradictions arise when Southern Gaza is shown to have higher hunger rates than Northern Gaza, which contradicts previous reports of more severe conditions in the north. Please support your results and expand on this topic.

Conclusion:

• The conclusion needs to be aligned with the rest of the paper, summarizing and validating the findings. It should not introduce new elements, like the cessation of hostilities, for the first time here.

References:

• Some references, such as those for ICRC protocols and the Global Nutrition Cluster, need corrections and consistency.

Images/Tables (if any):

• Tables 6-8 could be included as supplementary files, with only the significant p-values mentioned in the main text.

• Figures representing weight changes and hunger severity would improve clarity.

Language Quality:

• The overall language is clear and acceptable, though minor improvements could enhance readability.

Reviewer #4: Research Design and Rationale:

1. Clarity of Research Objectives:

o The abstract clearly states the aim to inform policymakers and humanitarian organizations, but could you clarify how the findings are specifically intended to inform policy or interventions in Gaza? Are there any particular groups or strategies you envision targeting with the findings?

2. Justification of Study Location:

o Why did you choose to conduct this study in Gaza specifically during this period of conflict? Could you elaborate on how the timing of the study (May to July 2024) may have influenced the findings, given that the situation in Gaza has been highly dynamic?

3. Previous Literature:

o How does your study compare to existing research on food insecurity in conflict zones, especially in Gaza? Are there any notable discrepancies with previous studies that report lower levels of food insecurity, and how do you explain these differences?

Sampling and Data Collection:

4. Sampling Strategy:

o The study surveyed 1209 households out of an initial 1503. Could you provide more details about the reasons behind excluding 294 households? Was this due to missing data, or were there other factors that may have affected the final sample?

5. Demographic Representation:

o You mention the mean age of participants is 38 years, with 53.5% being female. Were these demographics reflective of the overall Gaza population? Did you find any significant gender or age-based differences in food insecurity and hunger levels?

6. Data Collection in a Conflict Zone:

o Given the ongoing conflict, were there any particular challenges or limitations in collecting data in Gaza? For example, how did the security situation impact the study’s logistics or participants' willingness to engage?

7. Self-Reported Anthropometric Data:

o You collected self-reported anthropometric data, such as weight changes. How confident are you in the accuracy of these self-reports, considering the stress and trauma participants may be experiencing? Please insert this as limitations.

Interpretation of Results:

12. Hunger and Weight Loss:

• The average weight loss (from 74.6 kg to 64.1 kg) is striking. What do you believe are the key contributing factors to this weight loss beyond food insecurity, such as stress, healthcare access, or psychological impacts?

13. Statistical Impact of Confounding Variables:

• The abstract mentions that factors like displacement, age, economic status, and education significantly impacted hunger severity. Could you provide more details on the extent to which these variables influenced your findings? Were there any surprising or unexpected relationships?

Policy and Humanitarian Implications:

14. Recommendations for Action:

• Based on your findings, what specific recommendations would you propose to humanitarian organizations operating in Gaza to mitigate hunger and improve food security? Are there any quick-win interventions that could be implemented in the short term?

15. Long-term Policy Recommendations:

• Beyond immediate relief efforts, what longer-term policy changes do you think are necessary to address food insecurity in Gaza, particularly considering the ongoing conflict and blockade? Are there any impact on SDGs? Can you please indicate the importance of your study that shows the useless of international humanitarian rights and the SDGs.

6. PLOS authors have the option to publish the peer review history of their article (what does this mean? ). If published, this will include your full peer review and any attached files.

**Do you want your identity to be public for this peer review?** For information about this choice, including consent withdrawal, please see our Privacy Policy .

Reviewer #1: No

Reviewer #2: No

Reviewer #3: No

Reviewer #4: **Yes: ** Maha Hoteit

---

## [Author Response · Author response to Decision Letter 1]

27 Nov 2024

Responses to Reviewers’ Comments to the Author

Dear Editor,

PLOSONE

I am writing to submit our rebuttal and authors' response to the reviewers' comments regarding our manuscript, “Catastrophic Hinger in Gaza: Unprecedented Levels of Hunger Post-October 7th. A Real Population-Based Study from the Gaza Strip,” which we submitted on PLOSONE. We sincerely appreciate the reviewers' insights and constructive feedback, which have led to significant improvements in our work. In our response, we have addressed each of the reviewers' comments point-by-point and provided clarifications and modifications to enhance the clarity and robustness of our manuscript. These revisions have strengthened our arguments and contributed to a more comprehensive understanding of our findings.

We are grateful for the opportunity to address these comments and hope our revisions meet the reviewers' expectations.

I appreciate your consideration, and we look forward to your positive response.

Best regards,

Faris, ME

Corresponding author

Reviewer #1:

Comment: This paper is an important contribution to documenting Israel's genocide against Palestinians. While I appreciate the authors' attempts to use "neutral" language to describe and document the levels of food insecurity and hunger currently experienced by Palestinians, it is critical that the authors consider the connections between the hunger and famine being experienced in Gaza and the concept of starvation as a weapon of a genocidal war being perpetrated by Israel.

Response: Thanks for admiring our work, glad to hear. The use of Israeli forces for food and hunger as a weapon in their battle against the Palestinians has been documented and reported in our work; see lines 401-409.

Comment: I would therefore like to recommend that the authors consider the report produced by Human Rights Watch on this matter: https://www.hrw.org/news/2023/12/18/israel-starvation-used-weapon-war-gaza. The report asserts that (1) The Israeli government is using starvation of civilians as a method of warfare in the Gaza Strip, which is a war crime, and (2) Israeli officials have made public statements expressing their aim to deprive civilians in Gaza of food, water, and fuel – statements reflected in Israeli forces’ military operations.

Response: Thanks, add as suggested, please see lines 462-468.

Comment: On a personal note, as an author who has struggled to publish papers in peer-reviewed journals because reviewers tend to question the neutrality and objectivity of my analysis of Israel's genocidal tactics, I completely understand your decision and approach to this paper. Consequently, my comments above should be considered a recommendation rather than a prerequisite for publication.

Response: Thank you for your understanding. It is appreciated and means a lot to the authors. The authors will consider your comments a prerequisite.

Reviewer #2:

Comment: The manuscript is well written, the research study is of good quality, and the topic is of high importance in covering a pressing issue. Publications in this area are necessary. The researchers' access to Gaza could be improved, making the study very valuable.

Response: Thank you, we are glad to hear.

Some minor writing mistakes need to be corrected

Comment: On page 10, I suggest adding some information on the history of the blockade on Gaza.

Response: Added as suggested; see lines 422-439.

Comment: On page 12, the word "report" is repeated repeatedly, making this sentence redundant: "the report of the Global Nutrition Cluster (GNC) Report."

Response: Corrected; see lines 503-504.

Comment: Page 12 GNS should be GNC

Response: Corrected; see lines 505,509.

Comment: Page 15 Middle GA should be clarified

Response: Clarified as Middle Gaza; see line 654.

Reviewer #3:

Comment: General Comments: This paper addresses a highly relevant and urgent issue by examining hunger levels among Gaza residents after the military escalation on October 7th. The study aims to provide population-based data that could be valuable for informing policymakers and humanitarian organizations on addressing hunger and food insecurity.

Detailed Comments:

Title:

Comment: The title contains the word "Hunger, " which is redundant. A clearer, more concise alternative is "Catastrophic Hunger in Gaza: A Population-Based Study of Unprecedented Food Insecurity Post-October 7th".

Response: The title has been corrected and improved as per the suggestions, and famine and hunger are now included. The title becomes Catastrophic Famine in Gaza: Unprecedented Levels of Hunger Post-October 7th. A Real Population-Based Study from the Gaza Strip. Corrected; see the title on page 1, lines 1-2.

Abstract:

Comment: Background: The start is strong, but the ending mentions informing policymakers about strategies to mitigate hunger, which needs to be discussed in the paper. The purpose should instead emphasize providing insights into hunger levels and food insecurity rather than strategy.

Response: Corrected as suggested, please see lines 19-22, 41-44, 118-120, 380, 603, 606-613.

Comment: Methods: Sociodemographic data should be included in this section for completeness.

Response: Added, see lines 28-34.

Introduction:

Comment: The background of the Gaza crisis is well covered, but the section should include the most recent reports on hunger and malnutrition in Gaza.

Response: The most recent report of the UNGPH on October 2024 is added. See line 62-63.

Comment: Definitions of key terms like hunger, famine, starvation, and food security should be included to distinguish these closely related concepts.

Response: Added, se lines 76-94.

Comment: The introduction describes the Household Hunger Scale (HHS) in detail but merely mentions the Household Food Insecurity Access Scale (HFIAS) and the Household Food Security Survey Module (HFSSM). More balanced coverage is needed.

Response: The HHS text is significantly reduced to be comparable to the description of the other two tools, and the HFIAS and HFSSM are further described. A more balanced description can be found in lines 95-117.

Comment: It’s important to clarify that the study's purpose is to inform policymakers rather than to create strategies that align with the abstract.

Response: Corrected as suggested, please see lines 19-22, 41-44, 118-120, 380, 603, 606-613.

Materials and Methods:

Comment: This is a cross-sectional study conducted in the Gaza Strip from May to July 2024 during the ongoing war in the region. There is no clear description of how data were collected or whether the data collectors were trained.

Response: Description is added; see lines 131-135.

Comment: The method of data collection (e.g., paper-based, Google Docs) should be specified.

How the questionnaire was administered (online or in-person) needs to be clarified, though later in the results section, it is mentioned that households were interviewed.

Response: The same as before, the description has been added; see lines 131-135.

Comment: Participant eligibility criteria, informed consent processes, and an explanation of household hunger categories are absent.

Response: Informed consent is already there; see lines 137 and 145. Inclusion criteria are found in lines 141-145.

Comment: The explanation of the HFIAS tool could be made easier to understand through a table or appendix.

Response: The tool's explanation is transferred to the footnote of Table 4 for the HFIAS and removed from the main text. See Table 4, page 26. The reader is referred to the main reference [18].

Comment: The full questionnaire should be included in the appendix, and details about its development, validation, and scope (e.g., anthropometrics and sociodemographics) should be clearly outlined. It wasn’t clear how the questionnaire was built until we discovered that it combined the three scales with other questions.

Response: Added, see lines 152-157, 159-166.

Comment: More details are needed on how anthropometric data were collected and organized.

Response: The anthropometrics were self-reported, not measured (see the subheading, line 169). The validity and reliability of the self-reported anthropometric measurements are well documented in many published articles and systematically reviewed in one published article. Fayyaz et al. [26]; see lines 186-193.

Comment: The translation of assessment tools should mention who performed the translations and their qualifications.

Response: Added, see lines 152-154.

Comment: After viewing the data, a major flaw is the inconsistency in the data categories and geographical or current address names, which should be standardized to allow for proper analysis.

Response: The Gaza Strip is divided into three large governorates (prior to the recent Israeli reoccupation): the northern governorates (North Gaza, Gaza City), the middle (Deir Al Balah), and the southern (Khan Younis, Rafah). This governmental classification is consistent throughout the manuscript; see lines 130-131, 218-220, 325-330, 357-360, and 514.

Results:

Comment: Some data, such as economic status, should have been mentioned in the methods section but appear here for the first time. How was economic status categorized? What are the boundaries, and according to what?

Response: The respondents self-reported their economic status before the war; see lines 32, 162, 338, 366, and Table 1.

Comment: The presentation of results would benefit from visual aids like graphs or figures, particularly for illustrating shifts in BMI and weight loss.

Response: Done, added, see page 28.

Comment: The conclusion contradicts the results by stating that camp dwellers have higher hunger severity, which is not reflected in the results.

Response: This was described in Table 7 (which now becomes Supplementary Table 2), the question on Address before the war (City house/Home or Camp) and its relation to the level of household hunger.

Comment: When discussing the governorates, it is unclear whether the household location refers to where they are currently due to continuous displacement or where they originally came from.

Response: It was where they originally came from.

Comment: The phrase "with particular attention toward the three or more months by about one-third of the respondents" is unclear. How was this determined? Was it part of the questionnaire or a note from interviewers?

Response: The “three months” is mentioned in questions 5, 10, and 17 of the HFSSM (Table 3). The confusing statement has been rephrased for better clarity. See lines 271-273.

Comment: Results should be summarized clearly before diving into further detail.

Response: Done for the main five Tables (1-5) (the rest of Tables 6-8 are separated as Supplementary Tables 1-3, as indicated by one of the reviewers), see lines 215-218 (Table 1), 240-242 (Table 2), 260-265 (Table 3), 280-283 (Table 4), and 294-297 (Table 5).

Discussion:

Comment: The discussion is too lengthy and redundant in parts, especially when detailing the situation in Gaza before the war, which is not particularly related to the research aim. It should focus more on the study’s findings about hunger and food insecurity, backed by current evidence.

Response: The discussion on the nutritional status before the war is removed, and current evidence is added in this context. However, due to the extra new requirements suggested by the respected reviewers, the discussion becomes lengthier. More than ten new references were added and used in the discussion part.

Comment: The omission of study limitations, such as challenges faced during data collection in a conflict zone (e.g., safety, displacement, internet access, movement), especially in Northern Gaza, is significant. For example, the Rafah invasion and evacuation occurred at the beginning of the research period. How did this affect the research?

Response: Research limitations are expanded, and extra factors are added; see lines 615-637. Regarding the Rafah invasion and people's evacuation, people reached out in their newly arrived places, in their tents of camps, empty schools, or partially damaged houses, and were interviewed after being evacuated from Rafah.

Comment: More references are needed to strengthen some statements, and comparisons to similar studies would enhance the discussion.

Response: A comparison with relevant studies of famine and food insecurity amid military conflicts in Ethiopia and Nigeria has been added; see lines 398-400.

Comment: Non-participation challenges due to internet and war-related issues, such as the inability to reach people in tents either in the field or online, should be highlighted.

Response: Added, see lines 625-637.

Comment: The discussion should emphasize how the study’s results can inform humanitarian organizations in addressing hunger.

Response: Added; see lines 580-586, 613-614.

Comment: Contradictions arise when Southern Gaza is shown to have higher hunger rates than Northern Gaza, which contradicts previous reports of more severe conditions in the north. Please support your results and expand on this topic.

Response: Contradictions arise when Southern Gaza reports higher hunger rates than Northern Gaza, despite earlier assessments indicating more severe conditions in the north. This discrepancy can be understood through the geographical context and the original locations of displaced individuals. Historically, residents of Northern Gaza, especially in cities like Gaza City, have had a higher economic status and better access to income. This economic advantage has enabled many to retain savings, gold, or financial assistance, enhancing their ability to cope with crises and resulting in lower hunger prevalence. In contrast, individuals in Southern Gaza often lack similar resources, making them more susceptible to food insecurity. Additionally, many displaced people from Northern Gaza relocated to the Middle and Central areas during our data collection, further influencing hunger dynamics. These economic factors are crucial in understanding the varying hunger rates across different regions of Gaza; see lines 521-530.

Conclusion:

Comment: The conclusion needs to be aligned with the rest of the paper, summarizing and validating the findings. It should refrain from introducing new elements, like the cessation of hostilities, for the first time here.

Response: Please see lines 650-658.

References:

Comment: Some references, such as those for ICRC protocols and the Global Nutrition Cluster, need corrections and consistency.

Response: Corrected, thank you. Please check the reference numbers 40 and 41.

Images/Tables (if any):

Comment: Tables 6-8 could be included as supplementary files, with only the significant p-values mentioned in the main text.

Response: Done. Tables 6-8 are separated and included as supplementary files (Supplementary Table 1-2).

Comment: Figures representing weight changes and hunger severity would improve clarity.

Response: Changes in body weight and BMI are depicted in Figure 1, page 28. Hunger severity is a single value, not pre-post-war, and is reported in the tables for each assessment tool.

Comment: The overall language is clear and acceptable, though minor improvements could enhance readability.

Response: Thank you. The manuscript underwent comprehensive editing and linguistic revision. I hope all the minors are resolved.

Reviewer #4:

Research Design and Rationale:

Comment: The abstract clearly states that the findings aim to inform policymakers and humanitarian organizations, but could you clarify how they are specifically intended to inform policy or interventions in Gaza?

Response: Thank you for your insightful comment regarding the applicability of our findings to policymakers and humanitarian organizations. Due to the limited word count in the abstract, no further details can be added. This can be found more clearly in the text; see lines 41-43, 118-120, 607-614.

Comment: Are there any particular groups or strategies you envision targeting with the findings?

Response: Thank you for your question. Our findings will specifically target vulnerable populations, such as low-income families, pregnant and lactating women, ol

---

## [Decision Letter · Decision Letter 1]

5 Feb 2025

PONE-D-24-35303R1Catastrophic Famine in Gaza: Unprecedented Levels of Hunger Post-October 7th.  A Real Population-Based Study from the Gaza StripPLOS ONE

Dear Dr. Faris,

Thank you for submitting your manuscript to PLOS ONE. After careful consideration, we feel that it has merit but does not fully meet PLOS ONE’s publication criteria as it currently stands. Therefore, we invite you to submit a revised version of the manuscript that addresses the points raised during the review process.

We look forward to receiving your revised manuscript.

Kind regards,

Marianne Clemence, Staff Editor, on behalf of,

Mohammed Alkhaldi

Academic Editor

PLOS ONE

Journal Requirements:

Additional Editor Comments :

**Comments from the Journal Office:**

Thank you for your important submission highlighting the impact of the Gaza war on citizens. Following additional review by the staff editorial team, we are requesting a few revisions to the Discussion. PLOS One is primarily designed for the publication of scientific research, and our fourth publication criterion requires that “conclusions are presented in an appropriate fashion and are supported by the data” (https://journals.plos.org/plosone/s/criteria-for-publication). Whilst we have no concerns about the authors drawing attention to humanitarian concerns relating to the effects of war and blockade on food security, the results presented are not sufficient to support discussion regarding the interpretation of international law or political opinions. Inflammatory terminology, including in the keywords, should also be removed.

Therefore, we request that you please address the following as part of your revision. The manuscript will undergo further additional assessment by staff editors before the manuscript can be considered for publication. Therefore, please ensure that your revision is thorough, and that any new material is worded with care:

Provide additional appropriate references to support lines 424-434 "The current...food availability."Remove lines 402-403 "Our finding indicate...Israeli policies", the end of line 425 "In retaliation for this political choice", lines 469-470 "The current...war laws"" and 484-485 "The deliberate...humanitarian catastrophe", and any other content that speculates on the motivations of any person or institutionRemove lines 532-565 "The humanitarian crisis...population in Gaza" and any other content that analyzes the legality or legitimacy of any person or institutionRemove lines 665-666 "The international community...Palestinian people" and 668-670 "Additionally, advocating for...in the region", and ensure in particular that the Conclusions section is restricted to interpretations that are directly supported by the data presented in the results of the manuscript.Remove the following key words, and any other language that might be perceived as inflammatory, from your submission: Zionism, War criminals

Thank you very much for your attention to these requests, and thank you again for your submission to PLOS One.

Reviewers' comments:

Reviewer's Responses to Questions

**Comments to the Author**

1. If the authors have adequately addressed your comments raised in a previous round of review and you feel that this manuscript is now acceptable for publication, you may indicate that here to bypass the “Comments to the Author” section, enter your conflict of interest statement in the “Confidential to Editor” section, and submit your "Accept" recommendation.

Reviewer #4: All comments have been addressed

Reviewer #5: (No Response)

2. Is the manuscript technically sound, and do the data support the conclusions?

Reviewer #4: Yes

Reviewer #5: (No Response)

3. Has the statistical analysis been performed appropriately and rigorously? 

Reviewer #4: Yes

Reviewer #5: (No Response)

4. Have the authors made all data underlying the findings in their manuscript fully available?

Reviewer #4: Yes

Reviewer #5: (No Response)

5. Is the manuscript presented in an intelligible fashion and written in standard English?

Reviewer #4: Yes

Reviewer #5: (No Response)

6. Review Comments to the Author

Reviewer #4: Dear Authors I have reviewed your responses to the comments and concerns raised during the review process. You have adequately addressed all issues, including providing additional clarifications and making the necessary revisions to the manuscript. You have demonstrated a comprehensive understanding of the feedback and have strengthened the quality and reliability of the study.

I am satisfied with the responses, and I believe the article now meets the standards required for publication. I recommend the manuscript for acceptance and publication in its current form.

Reviewer #5: (No Response)

7. PLOS authors have the option to publish the peer review history of their article (what does this mean? ). If published, this will include your full peer review and any attached files.

**Do you want your identity to be public for this peer review?** For information about this choice, including consent withdrawal, please see our Privacy Policy .

Reviewer #4: **Yes: ** Maha Hoteit

Reviewer #5: **Yes: ** Dr. Haleama Al Sabbah

---

## [Author Response · Author response to Decision Letter 2]

26 Feb 2025

Editor, PLOS ONE

Public Library of Science

Subject: Revised Manuscript Submission: Catastrophic Famine in Gaza: Unprecedented Levels of Hunger Post-October 7th. A Real Population-Based Study from the Gaza Strip

Dear Editor,

Thank you for the opportunity to revise and resubmit our manuscript titled "Catastrophic Famine in Gaza: Unprecedented Levels of Hunger Post-October 7th. A Real Population-Based Study from the Gaza Strip." We appreciate the time and consideration given to our work by the editors and reviewers.

We have carefully addressed all the comments and suggestions raised during the review process. The revised manuscript is significantly strengthened as a result of the reviewers' insightful feedback. We have incorporated all of the requested changes, and we believe that the manuscript now meets the high standards of PLOS ONE.

We want to express our sincere gratitude to the reviewers for their constructive criticism and helpful suggestions. Their feedback has been invaluable in improving the quality, readability, and clarity of our manuscript.

This study provides crucial insights into the devastating impact of the recent conflict on food security and hunger in Gaza. The findings highlight the urgent need for humanitarian assistance and intervention to address the ongoing crisis. Our work will contribute to raising awareness of this critical issue and inform efforts to alleviate the suffering of the people in Gaza.

Thank you for your continued consideration of our manuscript.

We look forward to your final decision.

Sincerely,

MoezAlIslam,

moezalislma@gmail.com

Editors’ comments:

•Comment 1: Provide additional appropriate references to support lines 424-434, "The current...food availability."

•Response: Added, see references 29-32. Lines 486, 492, 495.

•Comment 2: Remove lines 402-403, "Our finding indicates...Israeli policies", the end of line 425, "In retaliation for this political choice," lines 469-470 "The current...war laws," and 484-485 "The deliberate...humanitarian catastrophe", and any other content that speculates on the motivations of any person or institution.

•Response: Done, removed.

•Comment 3: Remove lines 532-565, "The humanitarian crisis...population in Gaza," and any other content that analyzes the legality or legitimacy of any person or institution.

•Response: Done, removed.

•Comment 4: Remove lines 665-666, "The international community...Palestinian people" and 668-670, "Additionally, advocating for...in the region", and ensure in particular that the Conclusions section is restricted to interpretations that are directly supported by the data presented in the results of the manuscript.

•Response: Done, removed.

•Comment 5: Remove the following keywords and any other language that might be perceived as inflammatory, from your submission: Zionism, War criminals

•Response: Done, removed.

Reviewer’s comments:

Specific Comments

Abstract

•Comment 2: The abstract provides a concise summary but could better emphasize key findings and their implications.

•Response: Done, please see the abstract. Key findings are more emphasized.

•Comment 2: Revise the sentence on informing policymakers (lines 19–22) to focus on actionable insights rather than general aims.

•Response: Done, kindly see lines 22.

Introduction

•Comment 2: While the introduction offers comprehensive background information, it could be more concise, especially when discussing the historical context of food insecurity in Gaza.

•Response: The historical context of food insecurity in Gaza is reduced to be more concise; see lines 60-68.

•Comment 2: Definitions of key terms like hunger, food insecurity, and famine (lines 76–94) are helpful but should be condensed to improve clarity and avoid redundancy.

•Response: Done, please see lines 76-81.

Methods

•Comment 2: To enhance the reliability and validity of the methods, details on inclusion criteria (lines 141–145) and the translation process (lines 152–154) should be expanded.

•Response: Done, see lines 145-153, 160-172.

•Comment 2: Elaborate on the training process for data collectors (lines 131–135) to strengthen the methodological rigor.

•Response: Done, see lines 124-138.

Results

•Comment 2: The results are detailed, but incorporating visual aids such as graphs or charts would enhance clarity and engagement, particularly for shifts in BMI and hunger prevalence.

•Response: Added, please see Figures 1-3, pages 30-32.

•Comment 2: The categorization and impact of economic status on food insecurity (line 162) should be further elaborated.

•Response: Added, please see lines 185-202.

•Comment 2: Provide clearer categorization and contextual analysis of displacement data (line 130).

•Response: Added, see lines 203-214.

Discussion

•Comment 2: The discussion is thorough but occasionally redundant. Focus more on how the findings can directly inform policy and humanitarian interventions.

•Response: Please find in lines 604-618.

•Comment 2: The study limitations (lines 615–637) are well-addressed but should also include potential biases related to self-reported data.

•Response: Added, see lines 692-700.

Conclusion

•Comment 2: The conclusion is consistent with the findings but introduces new elements, such as the cessation of hostilities (line 650), which should be integrated earlier in the discussion.

•Response: Added, please see lines 602.

References

•Comment 2: Ensure consistency and accuracy in citations, particularly for references such as ICRC protocols (lines 40–41).

•Response: The paragraph including ICRC is removed based on one of the reviewer’s suggestions.

Figures and Tables

•Comment 2: Incorporate visual representations for key findings, such as weight loss and hunger severity, to improve the impact of the manuscript (e.g., BMI changes on Page 14).

•Response: Added, please see Figures 1-3, pages 30-32.

•

Page-Specific Comments

•Comment 2: Page 5, Line 62: Incorporate more recent reports on hunger to provide up-to-date context.

•Response: Not clear; page 5 includes lines 143-173, while line 62 is present on page 2. However, to meet this requirement, recent reports on hunger amid the armed conflicts in Sudan, Ethiopia, and Nigeria are added to provide up-to-date context. See lines 60-62.

•Comment 2: Page 10, Line 162: Clarify how economic status categories were defined and determined.

•Response: Added, please see lines 185-202.

•Comment 2: Page 14, Table 4: Simplify the table and key trends in the narrative.

•Response: Table 4 is presented on page 10, not 14. The table cannot be simplified more as it reflects the nine components of the assessment tool of HFIAS. In the narrative, key trends are presented; see lines 334-338.

•Comment 2: Page 18, Line 453: To contextualize the findings, compare them with similar studies on food insecurity in other conflict zones.

•Response: Added, see lines 462-464.

---

## [Decision Letter · Decision Letter 2]

2 Apr 2025

Catastrophic Famine in Gaza: Unprecedented Levels of Hunger Post-October 7th.  A Real Population-Based Study from the Gaza Strip

PONE-D-24-35303R2

Dear Dr. Faris,

We’re pleased to inform you that your manuscript has been judged scientifically suitable for publication and will be formally accepted for publication once it meets all outstanding technical requirements.

Kind regards,

Marianne Clemence, Staff Editor, on behalf of,

Mohammed Alkhaldi

Academic Editor

PLOS ONE

Additional Editor Comments (optional):

Reviewers' comments:

Reviewer's Responses to Questions

**Comments to the Author**

1. If the authors have adequately addressed your comments raised in a previous round of review and you feel that this manuscript is now acceptable for publication, you may indicate that here to bypass the “Comments to the Author” section, enter your conflict of interest statement in the “Confidential to Editor” section, and submit your "Accept" recommendation.

Reviewer #4: All comments have been addressed

Reviewer #5: All comments have been addressed

2. Is the manuscript technically sound, and do the data support the conclusions?

Reviewer #4: Yes

Reviewer #5: Yes

3. Has the statistical analysis been performed appropriately and rigorously? 

Reviewer #4: Yes

Reviewer #5: Yes

4. Have the authors made all data underlying the findings in their manuscript fully available?

Reviewer #4: Yes

Reviewer #5: Yes

5. Is the manuscript presented in an intelligible fashion and written in standard English?

Reviewer #4: Yes

Reviewer #5: Yes

6. Review Comments to the Author

Reviewer #4: The authors have addressed all comments, and the manuscript is now ready for publication.

All the comments raised were addressed

Reviewer #5: (No Response)

7. PLOS authors have the option to publish the peer review history of their article (what does this mean? ). If published, this will include your full peer review and any attached files.

**Do you want your identity to be public for this peer review?** For information about this choice, including consent withdrawal, please see our Privacy Policy .

Reviewer #4: **Yes: ** Maha Hoteit

Reviewer #5: **Yes: ** Dr. Haleama Al Sabbah

---

## [Editor Report · Acceptance letter]

PONE-D-24-35303R2

PLOS ONE

Dear Dr. Faris,

I'm pleased to inform you that your manuscript has been deemed suitable for publication in PLOS ONE. Congratulations! Your manuscript is now being handed over to our production team.

Kind regards,

on behalf of

Dr. Mohammed Alkhaldi

Academic Editor

PLOS ONE